# PREF-GRPO: PAIRWISE PREFERENCE REWARD-BASED GRPO FOR STABLE TEXT-TO-IMAGE REINFORCEMENT LEARNING

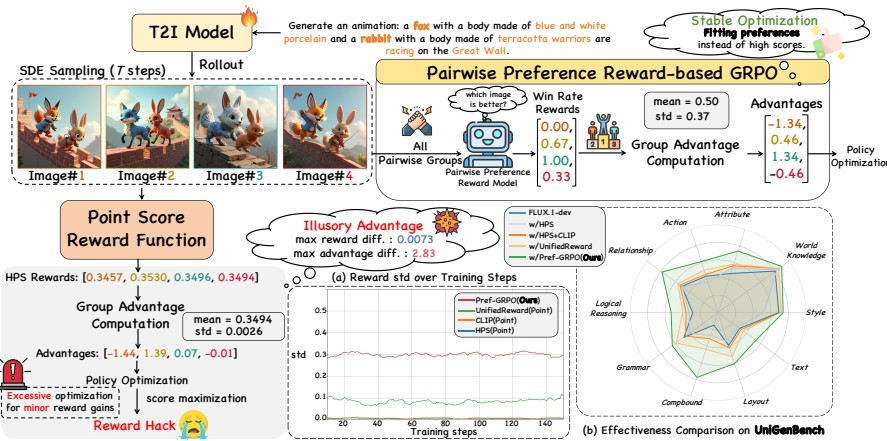

Figure 1: **Method Overview.** (a) Existing pointwise reward functions assign minimal score differences between generated images, which result in illusory advantage and ultimately lead to reward hacking. (b) PREF-GRPO shifts the training focus from reward score maximization to pairwise preference fitting, enabling stable optimization for T2I generation.

## ABSTRACT

Recent advancements underscore the significant role of GRPO-based reinforcement learning methods and comprehensive benchmarking in enhancing and evaluating text-to-image (T2I) generation. However, **(1)** current methods employ pointwise reward models (RM) to score a group of generated images and compute their advantages through score normalization for policy optimization. Although effective, this reward score-maximization paradigm is susceptible to **reward hacking**, where scores increase but image quality deteriorates. This work reveals that the underlying cause is illusory advantage, induced by minimal reward score differences between generated images. After group normalization, these small differences are disproportionately amplified, driving the model to over-optimize for trivial gains and ultimately destabilizing the generation process. To this end, this paper proposes **PREF-GRPO**, the first pairwise preference reward-based GRPO method for T2I generation, which shifts the optimization objective from traditional reward score maximization to pairwise preference fitting, establishing a more stable training paradigm. Specifically, in each step, the images within a generated group are pairwise compared using preference RM, and their win rate is calculated as the reward signal for policy optimization. Extensive experiments show that PREF-GRPO effectively differentiates subtle image quality differences, offering more stable advantages than pointwise scoring, thus mitigating the reward hacking problem. **(2)** Additionally, existing T2I benchmarks are **limited to coarse evaluation criteria**, covering only a narrow range of sub-dimensions and lacking fine-grained evaluation at the individual sub-dimension level, thereby hindering comprehensive

---

*Equal contribution. †Corresponding authors.

assessment of T2I models. Therefore, this paper proposes UNIGENBENCH, a unified T2I generation benchmark. Specifically, our benchmark comprises 600 prompts spanning 5 main prompt themes and 20 subthemes, designed to evaluate T2I models' semantic consistency across 10 primary and 27 sub evaluation criteria, with each prompt assessing multiple testpoints. Using the general world knowledge and fine-grained image understanding capabilities of Multi-modal Large Language Model (MLLM), we propose an effective pipeline for benchmark construction and evaluation. Through meticulous benchmarking of both open and closed-source T2I models, we uncover their strengths and weaknesses across various fine-grained aspects, and also demonstrate the effectiveness of our proposed PREF-GRPO.

# 1 INTRODUCTION

Recent progress highlights the pivotal importance of reinforcement learning (Liu et al., 2025; Li et al., 2025; Xue et al., 2025; He et al., 2025) and comprehensive benchmarking (Ghosh et al., 2023; Huang et al., 2023; Wei et al., 2025) in driving advancements and reliable evaluation of text-to-image (T2I) generation. Specifically, several GRPO-based approaches (Liu et al., 2025; Xue et al., 2025) employ pointwise reward models (RMs) (Wang et al., 2025b; Wu et al., 2023; Kirstain et al., 2023) to score a group of generated images in each step, followed by score normalization to compute advantages for policy optimization (Guo et al., 2025), which has proven highly effective in aligning T2I generation with human preferences. With these rapid developments, evaluating T2I models, particularly their instruction-following capability, has become a crucial challenge. Current widely adopted benchmarks (Huang et al., 2023; Ghosh et al., 2023), commonly assess T2I models by probing various compositional aspects and rely on CLIP (Radford et al., 2021) based metrics for quantitative evaluation. Recently, TIIF-Bench (Wei et al., 2025) has incorporated additional evaluation dimensions, such as text rendering, to provide a more comprehensive assessment.

Despite effectiveness, these studies encounter two key limitations: **(1)** Existing GRPO-based methods use pointwise RMs to achieve reward score maximization, which can provide early gains but often results in **reward hacking** (recognized by (Liu et al., 2025; Xue et al., 2025)) where scores increase but image quality deteriorates during continual learning, shown in Fig. 2. **(2)** Current T2I generation benchmarks provide **only primary dimension-level coarse evaluation**, covering a limited range of sub-dimensions and lacking fine-grained assessment at sub-dimension level, shown in Fig. 4.

In light of these issues, we posit that **(1)** reward hacking in GRPO-based methods stems from illusory advantage, which arises from the minimal score differences assigned by RMs between images within the group. When these scores are normalized into advantages, the small gaps are disproportionately amplified. Un-der a reward-maximization objective, such inflated advantages drive the policy to over-optimize for trivial reward cues, and this sustained pressure ultimately steers it toward reward-hacking behaviors that rapidly increase scores but destabilize the generation process (examples shown in Figs. 1 and 2). Besides, if the reward model is even slightly biased, this amplification magnifies these errors, driving the policy to exploit model flaws rather than align with human preferences. **(2)** The performance of current T2I models across most primary evaluation dimensions (*e.g.,* object at-

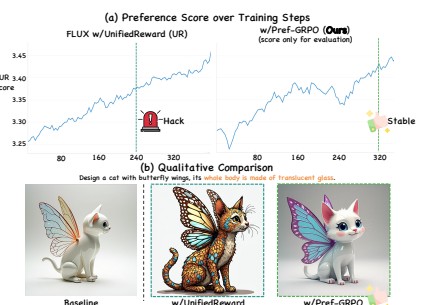

Figure 2: **Reward Hacking Visualization**.

tributes and actions) has reached a relatively high level, underscoring the necessity of decomposing these broad dimensions into finer-grained sub-tasks for more rigorous and comprehensive assessment.

To this end, this work proposes PREF-GRPO, the first preference reward-based GRPO method for stable T2I reinforcement learning, and UNIGENBENCH, a unified T2I generation benchmark for fine-grained semantic consistency evaluation. We elaborate on both in the following.

**(1) PREF-GRPO** incorporates a pairwise preference RM (PPRM) (Wang et al., 2025a), reformulating the GRPO optimization objective from conventional absolute reward score maximization to pairwise preference fitting. As illustrated in Fig. 1, in each step, given a set of generated images, we enumerate all possible image pairs and evaluate them with the PPRM to identify the preferred image in each

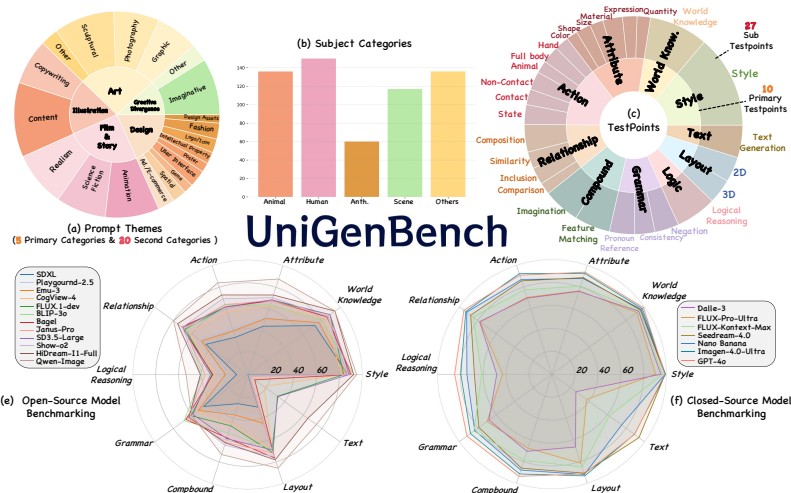

Figure 3: **Benchmark Statistics and Evaluation Results**. This figure presents (a) prompt themes, (b) subject distribution, and evaluation dimensions (testpoints) of UNIGENBENCH, along with benchmarking results for both open-source and closed-source T2I models.

| | Style | World Know. | Attribute | | | | | | Action | | | | | | Relationship | | | | Compound | | Grammar | | | Layout | | Logical Reason. | Text |
|---|---|---|---|---|---|---|---|---|---|---|---|---|---|---|---|---|---|---|---|---|---|---|---|---|---|---|---|
| | Score Mode | --- | --- | Quant. | Expn. | Material | Size | Shape | Color | Hand | Full Body | Animal | Non Contact | Contact | State | Compo. | Similarity | Inclusion | Comparison | Imagin. | Feaut"re Matching | Pronoun Ref. | Consistency | Negation | 2D | 3D | --- | --- |
| GenEval | Primary Dimension | | | ✓ | | | | | ✓ | | | | | | | | | | | | | | | | | | | |
| T2I-Comp | | | | ✓ | | | ✓ | ✓ | | | | | | | | | | | | | ✓ | | | | ✓ | ✓ | | |
| TIIF-Bench | | ✓ | ✓ | ✓ | | ✓ | ✓ | ✓ | ✓ | | ✓ | ✓ | ✓ | ✓ | | | | | | | ✓ | | ✓ | | ✓ | ✓ | | ✓ |
| **UniGenBench (Ours)** | **Primary &Sub Dimension** | ✓ | ✓ | ✓ | ✓ | ✓ | ✓ | ✓ | ✓ | ✓ | ✓ | ✓ | ✓ | ✓ | ✓ | ✓ | ✓ | ✓ | ✓ | ✓ | ✓ | ✓ | ✓ | ✓ | ✓ | ✓ | ✓ | ✓ |

Figure 4: **Benchmark Comparison**. While current methods only support scoring at the primary dimensions, our benchmark provides fine-grained evaluation across *both primary and sub dimensions*.

pair. The win rate of each image (computed as the proportion of pairwise comparisons it preferred) is then used as the reward signal for policy optimization. This design offers three key advantages: (a) **Amplified reward variance**: driving high-quality images toward win-rates near 1 and low-quality ones toward 0 yields more separable distributions and stable, informative advantage estimates. (b) **Enhanced robustness**: focus on relative rankings rather than absolute scores reduces over-optimization for marginal score gains and mitigates reward hacking. (c) **Preference alignment**: pairwise comparisons mirror human judgment for comparable images, producing reward signals that better capture nuanced preferences. Extensive experiments demonstrate that PREF-GRPO can discern subtle variations in image quality, yielding more stable and directional advantages than pointwise scoring, thereby enhancing optimization stability and alleviating reward hacking.

**(2)** Our **UNIGENBENCH** is built for fine-grained T2I evaluation, encompassing comprehensive evaluation dimensions, diverse prompt themes, and subject categories (see Fig. 3). Unlike existing benchmarks that provide only primary dimension-level coarse evaluation, most of our primary dimensions are further subdivided into fine-grained sub-dimensions (testpoints) shown in Fig. 4. We also construct an automated and effective pipeline based on the powerful Multi-modal Large Language Model (MLLM), *i.e.,* Gemini2.5-pro (Huang & Yang, 2025) for both benchmark construction and T2I model evaluation, as illustrated in Fig. 5. We benchmark popular closed-source models, including GPT-4o (Hurst et al., 2024), Nano Banana and Seedream-4.0 (Gao et al., 2025), as well as leading open-source models such as Qwen-Image (Wu et al., 2025a), Hidream (Cai et al., 2025), and Bagel (Deng et al., 2025). Our results, provided in Fig. 3 (e) and (f), show that both open- and closed-source models perform relatively well on prompts involving style and world knowledge, but consistently underperform on prompts requiring logical reasoning, such as those containing causal, contrastive, or other complex logical descriptions.

**Contributions**: (1) We present an analysis to reveal the fundamental cause of reward hacking as the illusory advantage problem. (2) Based on our analysis, we propose PREF-GRPO, the first pairwise preference reward-based GRPO method for stable T2I reinforcement learning, reformulating the optimization objective from conventional absolute reward score maximization to pairwise preference fitting. (3) Extensive experiments demonstrate that PREF-GRPO can discern subtle variations in image quality, producing more stable and directional advantages, thereby enhancing optimization

stability and alleviating reward hacking. (4) We introduce UNIGENBENCH, which encompasses comprehensive evaluation dimensions and diverse prompt themes, along with an effective pipeline for benchmark construction and T2I model evaluation. (5) Through meticulous evaluation of open- and closed-source T2I models, we reveal their strengths and weaknesses across various aspects.

## 2 RELATED WORK

**Reinforcement Learning for T2I Generation** is gaining rapid momentum. Early efforts pursued preference-driven objectives (Xie & Gong, 2025; Yang et al., 2024; Wallace et al., 2024). More recently, group relative policy optimization (GRPO) has advanced online RL-enhanced image generation. Flow-GRPO (Tong et al., 2025) and DanceGRPO (Xue et al., 2025) instantiate GRPO on flow-matching models, introducing stochasticity by recasting the original deterministic ODE as an equivalent SDE. While effective, these reward score-maximization methods are prone to reward hacking due to illusory advantage. To this end, we propose PREF-GRPO, which shifts training from reward-score maximization to pairwise preference fitting, yields more stable advantages, and thereby mitigates reward hacking.

**Existing Benchmark for T2I Evaluatoin** have expanded the evaluation of T2I models beyond simple visual fidelity, incorporating compositional reasoning (Ghosh et al., 2023; Huang et al., 2023) and world knowledge(Niu et al., 2025). Recently, (Wei et al., 2025) introduces TIIF-Bench, containing 5k prompts spanning multiple dimensions, *i.e.,* text rendering and style control, rigorously evaluating model robustness to variations in prompt length. However, existing benchmarks largely focus on primary dimension–level coarse assessment, covering a limited set of sub-tasks and lacking fine-grained assessment of these sub-tasks. To address this gap, we propose a unified image generation benchmark, UNIGENBENCH, consisting of 600 prompts spanning diverse themes and categories, assessing T2I models across 10 primary and 27 sub-criteria.

## 3 PREF-GRPO

This work introduces PREF-GRPO, aiming to establish a more stable RL paradigm for T2I generation, mitigating reward hacking in existing reward score-maximization GRPO methods. In this section, we first present the core idea of GRPO applied to flow matching models in Sec. 3.1, then analyze the root cause of reward hacking, *i.e.,* illusory advantage, in Sec. 3.2, and finally describe our proposed pairwise preference reward-based GRPO method in Sec. 3.3.

### 3.1 FLOW MATCHING GRPO

**Flow Matching.** Let $x_0 \sim \mathcal{X}_0$ be a data sample from the true distribution and $x_1 \sim \mathcal{X}_1$ a noise sample. Rectified flow (Liu et al., 2022) defines intermediate samples as

$$x_t = (1-t)x_0 + tx_1, \quad t \in [0,1], \tag{1}$$

and trains a velocity field $v_\theta(x_t, t)$ via the flow matching (Lipman et al., 2022) objective:

$$\mathcal{L}_{\text{FM}}(\theta) = \mathbb{E}_{t,x_0,x_1}\big[\|v - v_\theta(x_t,t)\|_2^2\big], \quad v = x_1 - x_0. \tag{2}$$

Beyond training, the iterative denoising process at inference time can be naturally formalized as a Markov Decision Process (Black et al., 2023). At each step $t$, the state is $s_t = (c, t, x_t)$, where $c$ denotes the prompt, and the action $a_t$ corresponds to producing the denoised sample $x_{t-1} \sim \pi_\theta(x_{t-1}|x_t, c)$. The transition is deterministic, *i.e.,* $s_{t+1} = (c, t-1, x_{t-1})$, with the initial state given by sampling a prompt $c \sim p(c)$, setting $t = T$, and drawing $x_T \sim \mathcal{N}(0, I)$. A reward is only provided at the final step: $R(x_0, c)$ if $t = 0$, and zero otherwise.

**GRPO on Flow Matching.** GRPO (Guo et al., 2025) introduces a group-relative advantage to stabilize policy updates. When applied to flow matching models, for a group of $G$ generated images $\{x_0^i\}_{i=1}^G$, the advantage of the $i$-th image is

$$\hat{A}_t^i = \frac{R(x_0^i, c) - \text{mean}(\{R(x_0^j, c)\}_{j=1}^G)}{\text{std}(\{R(x_0^j, c)\}_{j=1}^G)}. \tag{3}$$

The policy is updated by maximizing the regularized objective

$$\mathcal{J}_{\text{Flow-GRPO}}(\theta) = \mathbb{E}_{c,\{x^i\}}\Big[f(r, \hat{A}, \theta, \eta, \beta)\Big], \tag{4}$$

where

$$f(r, \hat{A}, \theta, \eta, \beta) = \frac{1}{G}\sum_{i=1}^{G}\frac{1}{T}\sum_{t=0}^{T-1}\min\big(r_t^i(\theta)\hat{A}_t^i, \text{clip}(r_t^i(\theta), 1-\eta, 1+\eta)\hat{A}_t^i\big) - \beta D_{\text{KL}}(\pi_\theta||\pi_{\text{ref}}), \tag{5}$$

with $r_t^i(\theta) = \frac{p_\theta(x_{t-1}^i|x_t^i,c)}{p_{\theta_{\text{old}}}(x_{t-1}^i|x_t^i,c)}$.

To satisfy GRPO's stochastic exploration requirements, (Liu et al., 2025) convert the deterministic Flow-ODE $dx_t = v_t dt$ to an equivalent SDE:

$$dx_t = \big(v_\theta(x_t, t) + \frac{\sigma_t^2}{2t}(x_t + (1-t)v_\theta(x_t, t))\big)dt + \sigma_t dw_t, \tag{6}$$

where $dw_t$ denotes Wiener process increments and $\sigma_t$ controls the stochasticity. Euler-Maruyama discretization gives the update rule:

$$x_{t+\Delta t} = x_t + \big(v_\theta(x_t, t) + \frac{\sigma_t^2}{2t}(x_t + (1-t)v_\theta(x_t, t))\big)\Delta t + \sigma_t\sqrt{\Delta t}\epsilon, \quad \epsilon \sim \mathcal{N}(0, I). \tag{7}$$

where $\sigma_t = a\sqrt{\frac{t}{1-t}}$ and $a$ is a scalar hyper-parameter that controls the noise level.

### 3.2 Illusory Advantage in Reward Score-Maximization GRPO Methods

Existing flow matching-based GRPO methods Liu et al. (2025); Xue et al. (2025); Li et al. (2025); He et al. (2025) use pointwise reward models (RMs) Wang et al. (2025b); Radford et al. (2021); Wu et al. (2023) to score a group of generated images in each training step. Then, the advantage of each generated image is computed by normalizing its reward score relative to the group, as shown in Eq. 3. This normalization standardizes the advantage across a group of samples. However, since existing pointwise RMs tend to assign overly similar reward scores $R(x_0^i, c)$ to comparable images within the same group, leading to an extremely small standard deviation $\sigma_r$. Consequently, the resulting normalized advantages can be excessively amplified (See example in Fig. 1). We refer to this phenomenon as *illusory advantage*.

Specifically, let $\mu_r$ denote the mean reward and $\sigma_r$ the standard deviation of the rewards in the group. When the rewards are close to each other, $\sigma_r \to 0$. In such cases, even a small difference $\Delta r = R(x_0^i, c) - \mu_r$ can lead to a large advantage:

$$\hat{A}_t^i = \frac{\Delta r}{\sigma_r}. \tag{8}$$

The disproportionate amplification of small reward differences, *i.e., illusory advantage*, has several detrimental effects: (1) **excessive optimization**: even minimal score variations are exaggerated, misleading the policy into over-updating and adopting extreme behaviors, *i.e.,* reward hacking (Fig. 2); (2) **sensitivity to reward noise**: the optimization becomes highly susceptible to biases or instabilities in the reward model, prompting the policy to exploit model flaws rather than align with true preferences.

### 3.3 Pairwise Preference Reward-based GRPO

To mitigate the *illusory advantage* problem in existing methods, we propose **PREF-GRPO**, which leverages a Pairwise Preference Reward Model (PPRM) (Wang et al., 2025a) to reformulate the optimization objective as pairwise preference fitting. Instead of relying on absolute reward scores, PREF-GRPO evaluates relative preferences among generated images, mirroring the human process of assessing two comparable images. This approach enables the reward signal to better capture nuanced differences in image quality, producing more stable and informative advantages for policy optimization while reducing susceptibility to reward hacking.

Specifically, given a set of $G$ images $\{x_0^i\}_{i=1}^G$ sampled from the policy $\pi_\theta$ for a prompt $c$, we enumerate all possible image pairs $(x_0^i, x_0^j)$ and use the PPRM to determine the preferred image in each pair. The *win rate* of image $i$ is defined as

$$w_i = \frac{1}{G-1} \sum_{j \neq i} \mathbb{1}\!\!\!\!\Vdash\!\left(x_0^i \succ x_0^j\right),\qquad(9)$$

where $\mathbb{1}\!\!\!\!\Vdash(\cdot)$ is the indicator function, and $x_0^i \succ x_0^j$ indicates that image $i$ is preferred over $j$ according to the PPRM. The win rates are then used as rewards for policy optimization, replacing the absolute rewards in the GRPO objective:

$$\hat{A}_t^i = \frac{w_i - \mathrm{mean}(\{w_j\}_{j=1}^G)}{\mathrm{std}(\{w_j\}_{j=1}^G)}.\qquad(10)$$

Compared to reward score maximization, Pref-GRPO offers several advantages: (1) **Amplified reward variance**: By transforming absolute reward scores into pairwise win-rates, Pref-GRPO inherently increases reward variance across a group of generated images. High-quality samples are pushed toward win-rates near 1, while lower-quality samples approach 0, producing a reward distribution that is both more discriminative and more robust for advantage estimation, thereby mitigating reward hacking. (2) **Robustness to reward noise**: Because the optimization relies on relative rankings rather than raw scores, Pref-GRPO substantially mitigates the amplified impact of small reward score fluctuations or biases in the reward model. This reduces the likelihood of the policy exploiting flaws in the reward signal, improving training stability. (3) **Alignment with human preference**: The pairwise formulation mirrors human perceptual evaluation. When comparing two images of similar quality, human judgments are inherently relative rather than absolute. By emulating this process, Pref-GRPO captures fine-grained quality distinctions often missed by pointwise scoring, providing a more faithful and reliable signal for policy improvement.

## 4 UNIGENBENCH

Existing benchmarks (Ghosh et al., 2023; Huang et al., 2023; Wei et al., 2025) exhibit following limitations: (1) **Limited coverage within coarse evaluation dimensions**: typically covering only a few sub-dimensions under each evaluation dimension, which fails to capture the full spectrum of model capabilities. For example, as shown in Fig. 4, current benchmarks include only a single sub-dimension for *relationships* and *grammar* dimensions, leading to an incomplete and potentially misleading assessment of model performance in these aspects. (2) **Absence of sub-dimension-level evaluation**: providing scores only at the primary evaluation dimension, without assessing individual sub-dimensions. This lack of granularity limits interpretability and hinders a detailed understanding of a T2I model's strengths and weaknesses.

Therefore, we propose **UNIGENBENCH**, a unified image generation benchmark that encompasses diverse prompt themes and a comprehensive set of fine-grained evaluation criteria. We will first introduce our design of prompt themes and evaluation criteria in the benchmark (Sec. 4.1), and then elaborate our MLLM-based automated pipeline for prompt generation and T2I evaluation (Sec. 4.2).

### 4.1 PROMPT THEME AND EVALUATION DIMENSIONS DESIGN

As shown in Fig. 3, UNIGENBENCH covers five major **prompt themes**: *Art*, *Illustration*, *Creative Divergence*, *Design*, and *Film&Storytelling*, further divided into 20 subcategories, alongside diverse **subject categories** including *animals*, *objects*, *anthropomorphic characters*, *scenes*, and an *Other* category for special entities (*e.g.,* robots in science-fiction themes). Unlike coarse metrics in existing benchmarks, we define 10 **primary evaluation dimensions** and 27 **sub-dimensions**, covering often overlooked aspects such as logical reasoning, facial expressions, and pronoun reference, enabling fine-grained evaluation and alignment with human intent. See Appendix B for more details.

### 4.2 BENCHMARK CONSTRUCTION AND EVALUATION PIPELINE

Having established diverse prompt themes, subject categories, and evaluation dimensions, we further construct an MLLM-based automated pipeline to operationalize the benchmark shown in Fig. 5. This

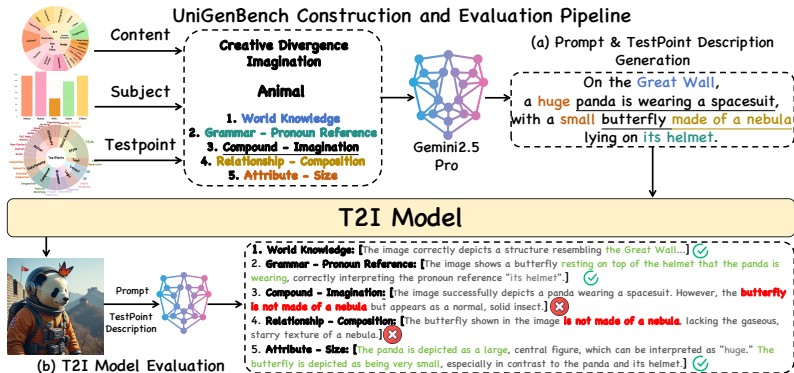

Figure 5: **UNIGENBENCH Construction and Evaluation Pipeline**. We leverage powerful MLLM for (a) large-scale and diverse prompts generation, and (b) scalable and fine-grained T2I evaluation.

pipeline serves two complementary purposes: (1) generating large-scale, diverse, and high-quality prompts in a systematic and controllable manner (Sec. 4.2.1), and (2) enabling scalable, reliable, and fine-grained evaluation of T2I models (Sec. 4.2.2). By leveraging the reasoning and perception capabilities of MLLMs, the pipeline eliminates the need for costly human annotation, while ensuring both efficiency and reliability in benchmark construction and model assessment.

### 4.2.1 PROMPT AND TESTPOINT DESCRIPTION GENERATION

Let $\mathcal{T}$ denote the set of prompt *themes*, $\mathcal{S}$ the set of *subject categories*, and $\mathcal{C}$ the set of *evaluation dimensions*. For each prompt, we sample a theme $t \sim \mathcal{T}$ and a subject category $s \sim \mathcal{S}$ uniformly at random. Subsequently, a subset of $k$ testpoints $\{c_1, \ldots, c_k\} \subset \mathcal{C}$, with $k \in [1, 5]$, is sampled to target specific fine-grained evaluation aspects.

The selected tuple $(t, s, \{c_1, \ldots, c_k\})$ is input into the MLLM, which generates two outputs: (i) a natural language prompt $p$ that conforms to the semantic constraints of the selected theme $t$ and subject category $s$, and (ii) a structured description set $\{d_1, \ldots, d_k\}$, where each $d_i$ specifies how the corresponding testpoint $c_i$ is realized in the prompt. Formally, this process can be expressed as:

$$(p, \{d_1, \ldots, d_k\}) \sim \text{MLLM}\Big(p, \{d_i\} \mid t, s, \{c_1, \ldots, c_k\}\Big). \tag{11}$$

### 4.2.2 T2I MODEL EVALUATION

Given the generated images $\{x_i\}$ for benchmark prompts $\{p_i\}$, we evaluate each image using an MLLM. Specifically, the image $x_i$, its corresponding prompt $p_i$, and its testpoint descriptions $\{d_{i,1}, \ldots, d_{i,k}\}$ are provided as input. The MLLM evaluates each testpoint $d_{i,j}$ in the context of $x_i$, producing a binary score $r_{i,j} \in \{0, 1\}$ and a textual rationale $e_{i,j}$ justifying the assessment. This can be formally represented as:

$$(r_{i,1}, \ldots, r_{i,k}, e_{i,1}, \ldots, e_{i,k}) \sim \text{MLLM}\Big(\{r_{i,j}, e_{i,j}\} \mid x_i, p_i, \{d_{i,1}, \ldots, d_{i,k}\}\Big). \tag{12}$$

This process ensures that the evaluation captures both the quantitative performance on each testpoint and the qualitative reasoning behind the assessment.

After obtaining the scores $r_{i,j}$ for each testpoint $d_{i,j}$ in all generated images, we aggregate them to compute scores of sub and primary evaluation dimensions. Specifically, for each sub-dimension $c$, we define its score as the ratio of the number of times the model successfully satisfies the corresponding testpoint description to the total number of occurrences of that testpoint across the benchmark:

$$R_c = \frac{\sum_{i,j} \mathbf{1}\{d_{i,j} \in c \text{ and } r_{i,j} = 1\}}{\sum_{i,j} \mathbf{1}\{d_{i,j} \in c\}}, \tag{13}$$

where $\mathbf{1}\{\cdot\}$ is the indicator function. The overall score for a primary dimension $C$ is then obtained by averaging the scores of all its sub-dimensions. This procedure ensures that both fine-grained performance on sub-dimensions and broader performance on primary dimensions are captured.

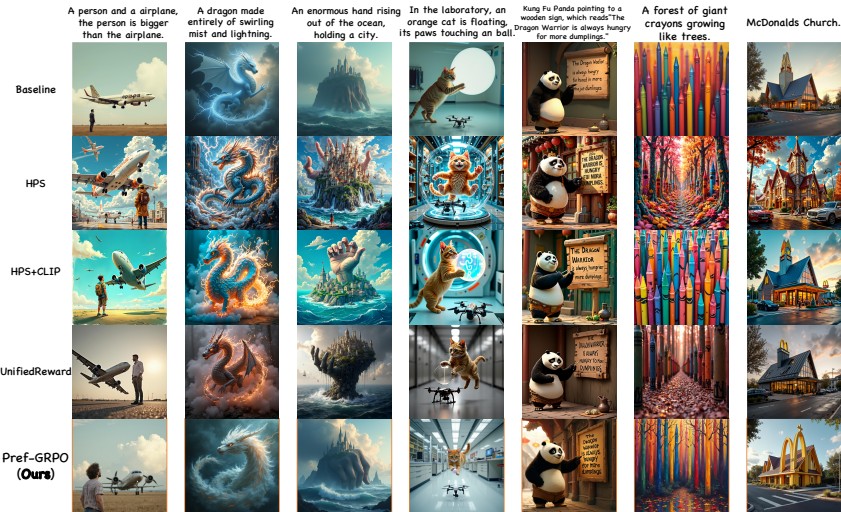

Figure 6: **Qualitative Comparison**. We compare PREF-GRPO with several pointwise RM-based GRPO methods, demonstrating its superior performance and effectiveness.

Table 1: **In-domain Semantic Consistency Comparison on UNIGENBENCH**. *Gemini2.5-pro* is used as the VLM for evaluation. Best scores are in **bold**, second-best in underlined.

| Model | Overall | Style | World Know. | Attribute | Action | Relation. | Logic.Reason. | Grammar | Compound | Layout | Text |
|---|---|---|---|---|---|---|---|---|---|---|---|
| FLUX.1-dev | 61.30 | 83.90 | 88.92 | 67.84 | 62.17 | 67.26 | 30.91 | 60.96 | 47.04 | 71.83 | 32.18 |
| w/ HPS | 58.77 | 75.20 | 88.77 | 66.56 | 58.94 | 66.88 | 28.18 | 58.02 | 45.88 | 67.91 | 31.32 |
| w/ HPS&CLIP | 61.81 | 84.92 | 88.98 | 68.44 | 62.54 | 68.10 | 31.01 | 59.36 | 50.60 | 71.07 | 33.07 |
| w/ UnifiedReward | 63.62 | 86.10 | 89.72 | 71.55 | 63.69 | 70.42 | 32.05 | 62.43 | 52.32 | 73.51 | 34.44 |
| **FLUX+Pref-GRPO** | **69.46** | **88.40** | **90.35** | **75.00** | **69.77** | **76.52** | **44.09** | **63.27** | **62.43** | **77.61** | **47.13** |

## 5 EXPERIMENT

### 5.1 IMPLEMENTATION DETAILS

**Baselines**: We use FLUX.1-dev (Labs., 2024) as base model and UnifiedReward-Think (Wang et al., 2025a) for pairwise preference RM in PREF-GRPO. For reward-maximization baseline comparison, we employ HPS (Wu et al., 2023), CLIP (Radford et al., 2021), and UnifiedReward (UR) (Wang et al., 2025b). **Training and Evaluation**: We generate 5k prompts using our pipeline (Fig. 5(a)) for training and evaluate models on UNIGENBENCH. Each test prompt generates four outputs for evaluation. Out-of-domain semantic consistency is assessed with GenEval (Ghosh et al., 2023) and T2I-CompBench (Huang et al., 2023), while image quality is evaluated using UR (Wang et al., 2025b), ImageReward (Xu et al., 2023), PickScore (Kirstain et al., 2023), and Aesthetic (Schuhmann., 2022).

### 5.2 RESULTS OF PREF-GRPO

**Quantitative.** As shown in Tabs. 1 and 2, our PREF-GRPO demonstrates substantial improvements in both semantic consistency and image quality. For example, on UNIGENBENCH, relative to UR-based score-maximization approaches, Pref-GRPO attains a 5.84% increase in the *overall* score, with further improvements of 12.69% on *Text* and 12.04% on *Logical Reasoning*. In image quality evaluation, our method also achieves comprehensive advantages. **Qualitative.** Examples are shown in Fig. 6. Notably, existing methods exhibit varying degrees of reward hacking. For instance, HPS-optimized images tend to be oversaturated, while UR-optimized images appear darker. We also explore mitigating reward hacking by combining multiple reward scores, *i.e.,* using HPS+CLIP jointly (third row in Fig. 6). While this reduces reward hacking, it does not fully resolve the issue. In contrast, our method mitigates reward hacking while markedly improving semantic generation. **Reward Hacking Analysis.** We visualize the evolution of image quality scores during training for both UR-based score-maximization methods and PREF-GRPO. As shown in Fig. 2, while UR-based models exhibit rapid score increases, inspection of intermediate results reveals a degradation in actual image quality. In contrast, our Pref-GRPO, though fitting pairwise preferences and yielding relatively

Table 2: **Out-of-Domain Semantic Consistency and Image Quality Evaluations**. The best results are in **bold**, and the second best are underlined.

| Model | Semantic Consistency | | | Image Quality | | | |
|---|---|---|---|---|---|---|---|
| | UniGenBench | T2I-CompBench | GenEval | UnifiedReward | PickScore | ImageReward | Aesthetic |
| FLUX.1-dev | 61.30 | 48.17 | 62.92 | 3.04 | 22.42 | 1.27 | 6.13 |
| w/ HPS | 58.77 | 46.77 | 59.31 | 3.09 | 22.62 | 1.34 | 6.20 |
| w/ HPS+CLIP | 61.81 | 49.18 | 64.85 | 3.08 | 22.61 | 1.30 | 6.25 |
| w/ UnifiedReward | 63.62 | 50.20 | 67.28 | 3.14 | 22.88 | 1.38 | 6.31 |
| **FLUX+Pref-GRPO** | **69.46** | **51.85** | **70.53** | **3.26** | **23.02** | **1.44** | **6.52** |

Table 3: **Benchmarking Results of T2I models on UNIGENBENCH**. *Gemini2.5-pro* is used as the VLM for evaluation. Best scores are in **bold**, second-best in underlined.

| Model | Overall | Style | World Know. | Attribute | Action | Relation. | Logic.Reason. | Grammar | Compound | Layout | Text |
|---|---|---|---|---|---|---|---|---|---|---|---|
| | | | | *Closed-source Models* | | | | | | | |
| Keling-Ketu | 65.93 | 92.27 | 86.62 | 71.66 | 68.73 | 70.94 | 43.75 | 71.26 | 60.81 | 77.23 | 16.03 |
| DALL-E-3 | 69.18 | 95.06 | 93.51 | 75.97 | 69.83 | 78.06 | 48.18 | 68.07 | 70.60 | 66.67 | 25.86 |
| FLUX-Pro-Ultra | 70.67 | 90.60 | 91.61 | 76.50 | 70.53 | 77.54 | 43.18 | 70.05 | 67.78 | 81.53 | 37.36 |
| Seedream-3.0 | 78.95 | 98.10 | 95.25 | 85.58 | 82.98 | 80.84 | 52.73 | 61.36 | 73.84 | 87.31 | 71.55 |
| FLUX-Kontext-Max | 80.00 | 96.59 | 94.19 | 80.93 | 77.38 | 85.08 | 61.36 | 84.23 | 78.99 | 85.04 | 61.92 |
| Seedream-4.0 | 87.35 | 98.80 | 95.41 | 88.57 | 85.65 | 87.69 | 67.73 | 78.88 | 86.08 | 90.67 | **93.97** |
| 🥉 Nano Banana | 87.45 | 98.87 | 96.32 | 87.84 | 86.83 | 92.00 | 74.26 | 83.36 | 87.83 | 91.96 | 75.22 |
| 🥈 Imagen-4.0-Ultra | 91.54 | **99.20** | 97.47 | 92.52 | **92.20** | 93.02 | 79.55 | **87.97** | 91.37 | **93.10** | 89.08 |
| 🥇 GPT-4o | **92.77** | 98.57 | **98.87** | **93.59** | 90.79 | **94.97** | **84.97** | 91.76 | **93.55** | 91.35 | 89.24 |
| | | | | *Open-source Models* | | | | | | | |
| SDXL | 39.75 | 87.40 | 72.63 | 44.34 | 34.22 | 44.92 | 9.55 | 47.33 | 26.68 | 29.85 | 0.57 |
| Playground 2.5 | 45.61 | 89.50 | 76.11 | 52.78 | 42.68 | 51.52 | 16.59 | 53.21 | 35.44 | 37.13 | 1.15 |
| Emu3 | 46.02 | 86.80 | 77.06 | 51.39 | 40.11 | 49.75 | 19.32 | 52.94 | 36.86 | 44.78 | 1.15 |
| Janus-flow | 46.39 | 86.20 | 62.50 | 47.97 | 43.35 | 50.00 | 21.14 | 60.29 | 45.10 | 46.46 | 0.86 |
| Janus | 51.23 | 89.90 | 73.58 | 54.81 | 50.38 | 55.08 | 26.82 | 59.09 | 46.65 | 54.85 | 1.15 |
| Hunyuan-DiT | 51.38 | 94.10 | 80.70 | 62.71 | 49.05 | 59.64 | 24.55 | 55.48 | 41.62 | 44.78 | 1.15 |
| CogView4 | 56.30 | 82.00 | 83.07 | 63.25 | 57.51 | 62.44 | 28.18 | 54.81 | 44.72 | 69.22 | 17.82 |
| BLIP3-o | 59.87 | 92.80 | 80.22 | 63.89 | 63.97 | 66.50 | 39.55 | **68.45** | 53.74 | 68.47 | 1.15 |
| FLUX.1-dev | 61.30 | 83.90 | 88.92 | 67.84 | 62.17 | 67.26 | 30.91 | 60.96 | 47.04 | 71.83 | 32.18 |
| Bagel | 61.53 | 90.20 | 85.60 | 67.74 | 61.98 | 70.69 | 30.23 | 66.44 | 58.12 | 76.49 | 7.76 |
| Janus-Pro | 61.61 | 90.80 | 86.71 | 67.74 | 64.26 | 68.40 | 37.05 | 64.44 | 62.11 | 72.01 | 2.59 |
| Show-o2 | 62.73 | 87.20 | 86.08 | 70.51 | 69.58 | 70.18 | 40.91 | 61.63 | 64.69 | 75.37 | 1.15 |
| SD-3.5-Large | 62.99 | 88.60 | 88.92 | 68.59 | 62.17 | 69.80 | 32.27 | 58.96 | 58.76 | 69.03 | 32.76 |
| 🥉 Pref-GRPO | 69.46 | 88.40 | 90.35 | 75.00 | 69.77 | 76.52 | 44.09 | 63.27 | 62.43 | 77.61 | 47.13 |
| 🥈 HiDream-I1-Full | 71.81 | 92.50 | 94.15 | 72.97 | 73.00 | 75.38 | 41.14 | 63.24 | 62.63 | 78.17 | 64.94 |
| 🥇 Qwen-Image | **78.81** | **95.10** | **94.30** | **87.61** | **84.13** | **79.70** | **53.64** | 60.29 | **73.32** | **85.82** | **74.14** |

more gradual score growth, demonstrates consistent and stable improvements in visual quality and effectively mitigates reward hacking. See Appendix A.4 for more analyses.

## 5.3 BENCHMARKING RESULTS ON UNIGENBENCH

As shown in Tab. 3, closed-source models deliver the strongest results: GPT-4o Hurst et al. (2024) and Imagen-4.0-Ultra Saharia et al. (2022) lead across most dimensions, particularly *logical reasoning*, *text rendering*, *relationship understanding*, and *compound*, indicating robust semantic alignment and understanding. Open-source models are improving: Qwen-Image (Wu et al., 2025a) and HiDream (Cai et al., 2025) consistently rank at the top among open models, with notable strengths in *Action*, *layout*, and *attribute*, narrowing the gap with closed-sourced models. Despite this progress, limitations still remain. Most open- and closed-source models have not yet reached saturation on the most challenging dimensions, particularly *logical reasoning* and *text rendering*, leaving substantial room for improvement. Moreover, open-source models tend to exhibit greater instability across dimensions, often lagging in *grammar* and *compound* tasks. See Appendix B.2 for sub-dimension-level evaluation.

## 6 CONCLUSION

We propose PREF-GRPO, the first pairwise preference reward-based GRPO method, offering a more stable T2I reinforcement learning paradigm. Besides, we introduce UNIGENBENCH, a unified T2I generation benchmark that encompasses comprehensive dimensions and diverse prompt themes. Extensive experiments validate the effectiveness of our method and the reliability of the benchmark.

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

# A  PREF-GRPO

## A.1  WHY PAIRWISE PREFERENCE-BASED REWARD WORKS

This work finds that reward hacking is fundamentally caused by the model overly aligning with the reward model's preferences. Specifically, we observe that HPS tends to favor images with saturated colors. However, when the model excessively optimizes this preference, reward hacking occurs, resulting in extreme saturation across all generated images. In contrast, stable optimization should yield subtle adjustments, such as moderately bright colors.

Existing works (Liu et al., 2025; Xue et al., 2025) also discuss the issue of reward hacking, recognizing it as a pervasive challenge in the field. However, these methods typically attempt to alleviate the problem by adjusting experimental settings, such as incorporating the KL loss (Liu et al., 2025). In contrast, our work reveals that the underlying cause of reward hacking is the issue of illusory advantage. This drives the model to continually over-optimize for trivial reward score improvements, exacerbating reward hacking.

Some trivial methods, such as directly scaling down the advantage or scaling up the standard deviation during reward normalization, although they mitigate illusory advantage to some extent, come at the cost of reduced model learning ability. Scaling down the advantage essentially reduces the learning rate by limiting the magnitude of updates. Scaling up the standard deviation dampens the model's ability to distinguish meaningful reward differences, leading to a less responsive learning process.

In contrast, our work explores a more stable reward mechanism: pairwise preference fitting. We analyze the rationale behind this mechanism as follows: (1) During GRPO, reward models act as proxies for human judgment, guiding the model's training process. However, human evaluators typically make relative comparisons between comparable images, rather than assigning absolute scores to each image. This relative comparison allows for a more accurate capture of subtle differences in image quality, ensuring that the model better aligns with human preferences. (2) Moreover, even when occasional errors occur in pairwise preference-based rewards, these errors are not amplified in the same way as errors in reward score maximization learning. This is because pairwise preference fitting provides more stable advantages, ensuring that small errors do not disproportionately influence the optimization process. As a result, the model's training process is less prone to destabilization.

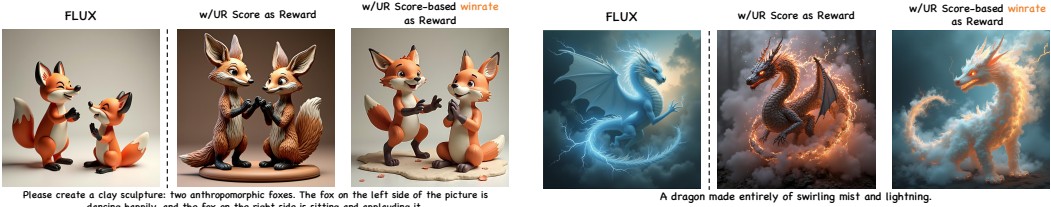

Figure 7: **Qualitative Results of UnifiedReward Score-based Winrate as Reward**: We convert UnifiedReward scores into win rates as rewards for GRPO, and observe that this effectively mitigates the reward hacking issue.

## A.2  POINT SCORE-BASED WINRATE V.S. PAIRWISE PREFERENCE-BASED WINRATE

To demonstrate that shifting the training objective to our proposed pairwise preference fitting enhances stability and that pairwise comparisons are more reliable than pointwise scores, we conduct an experiment using point score-based win rates as rewards. Specifically, we convert the UnifiedReward scores for a group of images into win rates by comparing the scores of each image pair. This win rate then serves as the reward signal for training. As shown in Fig. 7, when the training objective shifts to pairwise preference fitting, the previously dominant dark style in the images is notably alleviated, which validates that pairwise preference fitting stabilizes training. We also provide quantitative results in Tab. 4, demonstrating that although using point score-based win rates as rewards yields significant improvements over reward score maximization, our method using pairwise preference rewards achieves even better results. This confirms that relative comparisons between images are more reliable than absolute point-based scoring.

Table 4: **Exploration of Sampling Steps and Joint Optimization**. The best results are in **bold**, and the second best are underlined.

| Model | Semantic Consistency | | Image Quality | | | |
| --- | --- | --- | --- | --- | --- | --- |
| | UniGenBench | GenEval | UnifiedReward | PickScore | ImageReward | Aesthetic |
| FLUX.1-dev | 61.30 | 62.92 | 3.04 | 22.42 | 1.27 | 6.13 |
| *Point Score-based Winrate v.s. Pairwise Preference-based Winrate* | | | | | | |
| FLUX+UR (score) | 63.62 | 67.28 | 3.14 | 22.88 | 1.38 | 6.31 |
| FLUX+UR (winrate) | 64.32 | 68.13 | 3.20 | 22.91 | 1.39 | 6.35 |
| **Pref-GRPO** | **69.46** | **70.53** | **3.26** | **23.02** | **1.44** | **6.52** |
| *Exploration of **Sampling Steps** during Rollout* | | | | | | |
| Pref-GRPO w/ 16 steps | 68.12 | 67.99 | 3.12 | 22.89 | 1.36 | 6.33 |
| w/ 20 steps | 69.23 | 68.92 | 3.18 | 22.94 | **1.48** | 6.43 |
| w/ **25 steps** | 69.46 | **70.53** | **3.26** | **23.02** | 1.44 | **6.52** |
| w/ 30 steps | **69.49** | 70.51 | 3.22 | 22.97 | 1.46 | 6.48 |
| *Join Optimization of Pref-GRPO and Reward Score-Maximization* | | | | | | |
| **Pref-GRPO** | 69.46 | 70.53 | **3.26** | **23.02** | **1.44** | **6.52** |
| **Pref-GRPO+CLIP** | **70.02** | **71.26** | 3.18 | 22.86 | 1.41 | 6.44 |

## A.3 MORE IMPLEMENTATION DETAILS

Training is conducted on 64 H20 GPUs with 25 sampling steps, 8 rollouts per prompt from the same initial noise, 4 gradient accumulation steps, and a learning rate of $1 \times 10^{-5}$. Following (Liu et al., 2025), we set the hyperparameter $a = 0.7$. We deploy pairwise preference reward server via vLLM (Kwon et al., 2023). For inference, we adopt 30 sampling steps and a classifier-free guidance scale of 3.5, consistent with the official Flux configuration.

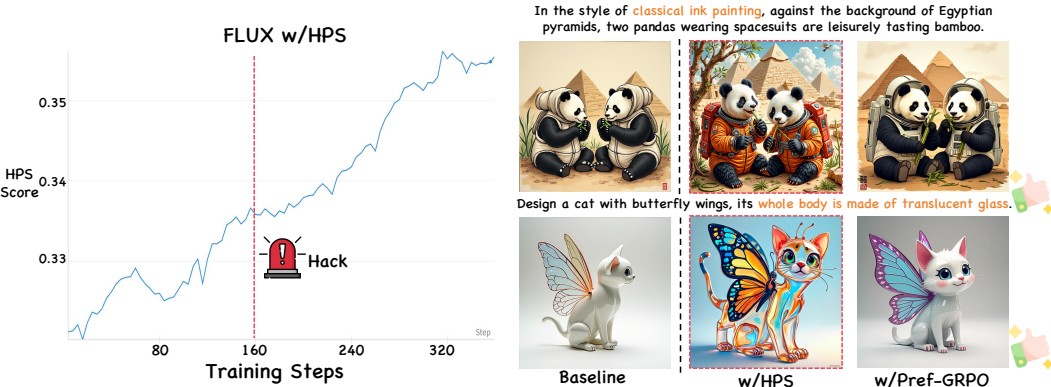

Figure 8: **Reward Hacking Visualization of HPS**. At around step 160, the image quality begins to degrade, even though the reward score continues to rise, indicating the occurrence of reward hacking.

## A.4 MORE REWARD HACKING ANALYSIS

We further visualize the phenomenon of reward hacking when using HPS (Wu et al., 2023) as the pointwise reward model. As shown in Fig. 8, the reward score increases sharply during training, yet the model quality begins to deteriorate around step 160, manifesting as over-saturated colors. Despite this degradation, the reward score continues to rise. This indicates the presence of the *illusory advantage* problem, where the model excessively optimizes for marginal improvements in reward. Under prolonged pressure, the model deviates towards a hacked trajectory that rapidly inflates reward scores while compromising generation quality.

Additionally, we observe that HPS exhibits reward hacking more rapidly compared to UnifiedReward. This is likely because HPS assigns even more minimal reward score differences between generated images, resulting in a smaller standard deviation, as shown in Fig. 1, which exacerbates the issue of illusory advantage.

Table 5: **Out-of-Domain Performance Comparison on GenEval**. The best results are in **bold**, and the second best are underlined.

| Model | Overall | Single Obj. | Two Obj. | Counting | Colors | Position | Attr. Binding |
|---|---|---|---|---|---|---|---|
| FLUX.1-dev | 62.92 | 97.81 | 79.55 | 71.56 | 77.66 | 18.50 | 42.25 |
| w/ HPS | 59.31 | 97.43 | 75.00 | 62.81 | 73.67 | 21.00 | 34.75 |
| w/ HPS+CLIP | 64.85 | 98.12 | 81.00 | 71.81 | 78.44 | 19.00 | 40.75 |
| w/ UnifiedReward | 67.28 | 98.43 | 82.57 | 72.25 | 79.72 | 21.25 | 49.50 |
| **FLUX+Pref-GRPO** | **70.53** | **99.38** | **86.36** | **74.06** | **81.12** | **26.00** | **57.25** |

Table 6: **Out-of-Domain Performance Comparison on T2I-CompBench.** The best results are in **bold**, and the second best are underlined.

| Model | Overall | Color | Shape | Texture | 2D-Spatial | 3D-Spatial | Numeracy | Non-Spatial | Complex |
|---|---|---|---|---|---|---|---|---|---|
| FLUX.1-dev | 48.17 | 77.34 | 48.32 | 62.66 | 28.01 | 40.04 | 61.88 | 30.67 | 36.49 |
| w/ HPS | 46.77 | 78.17 | 51.55 | 66.13 | 22.06 | 33.75 | 56.34 | 30.20 | 35.96 |
| w/ HPS+CLIP | 49.18 | 78.44 | 53.22 | 64.24 | 26.90 | 40.83 | 61.58 | 30.56 | 37.69 |
| w/ UnifiedReward | 50.20 | 78.32 | 55.13 | 67.44 | 28.91 | 40.04 | 62.47 | 30.88 | 38.39 |
| **FLUX+Pref-GRPO** | **51.85** | **80.27** | **56.01** | **69.12** | **28.93** | **43.95** | **65.92** | **31.05** | **39.58** |

### A.5 OUT-OF-DOMAIN SEMANTIC EVALUATION

We provide detailed out-of-domain semantic generation evaluations in Tabs. 5 and 6, which highlight the notable improvements of our method compared with existing approaches.

### A.6 SAMPLING STEPS ANALYSIS

We further investigate the impact of the number of sampling steps during rollout on both semantic consistency and image quality. As shown in Tab. 4, increasing the sampling steps from 16 to 25 consistently improves performance across all metrics, with the best results achieved at 25 steps. Although 30 steps yield comparable results to 25, the additional computation brings higher time costs without clear gains. Therefore, we adopt 25 sampling steps as the default setting, which strikes the best balance between effectiveness and efficiency.

### A.7 JOINT OPTIMIZATION OF PAIRWISE PREFERENCE FITTING AND REWARD SCORE MAXIMIZATION

Although reward score maximization inherently risks reward hacking, we hypothesize that incorporating our pairwise preference fitting mechanism for joint optimization can substantially mitigate this issue. To validate this, we conduct joint optimization using a simple yet effective reward model, *i.e.,* CLIP (Radford et al., 2021). As shown in Tab. 4, the integration of CLIP notably improves semantic consistency, but this gain comes at the expense of slightly reduced image quality, highlighting a trade-off between semantic alignment and visual fidelity. We also provide qualitative comparison results in Fig. 9, where, despite the quality trade-off, no reward hacking phenomenon is observed. These results indicate that pairwise preference fitting acts as a regularizer when combined with reward score maximization, providing a principled way to balance semantic accuracy and visual quality while mitigating reward hacking.

## B UNIGENBENCH

### B.1 BENCHMARKING MODELS

**Closed-source Models.** GPT-4o (Hurst et al., 2024), Imagen3.0/4.0-ultra Saharia et al. (2022), Seedream-3.0 (Gao et al., 2025), DALL-E-3 (OpenAI), FLUX-Pro-Ultra/Kontext-Max (Labs., 2024), and Keling-Ketu (Kuaishou).

**Open-source Models.** Qwen-Image (Wu et al., 2025a), Hidream (Cai et al., 2025), Show-o2 (Xie et al., 2025), SD-3.5-Large (Rombach et al., 2021), Janus-Pro (Chen et al., 2025b), Flux.1-dev (Labs.,

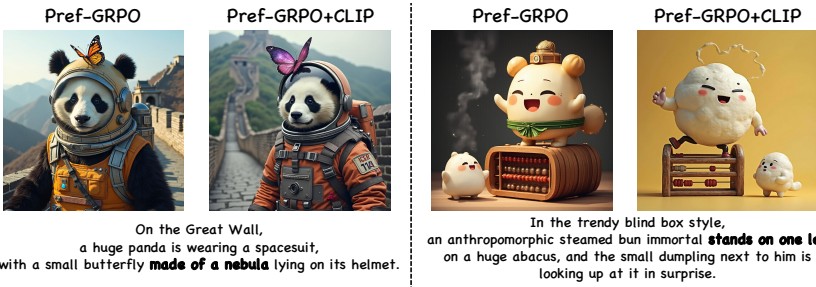

Figure 9: **Qualitative Results of Joint Optimization**. Joint training with CLIP improves semantic consistency while slightly degrading perceptual quality, reflecting the inherent trade-off.

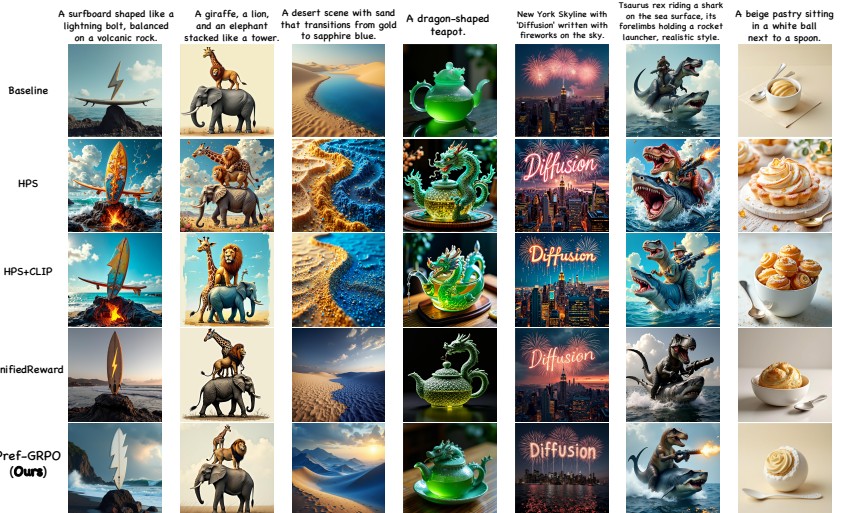

Figure 10: **More Qualitative Comparison**. We compare PREF-GRPO with several pointwise RM-based GRPO methods, demonstrating its superior performance and effectiveness.

2024), Bagel (Deng et al., 2025), BLIP3-o (Chen et al., 2025a), CogVideo4 (Ding et al., 2021), Hunyuan-DiT (Li et al., 2024b), Janus (Wu et al., 2025b), Janus-flow (Ma et al., 2024), Emu3 (Wang et al., 2024), Playground2.5 (Li et al., 2024a), and SDXL (Rombach et al., 2021).

## B.2  FINE-GRAINED EVALUATION RESULTS

Existing benchmarks are limited to evaluating only primary dimensions, without capturing the performance of models on more granular aspects. In contrast, our UNIGENBENCH enables fine-grained assessment across both primary dimensions and their corresponding sub-dimensions. The detailed evaluation results are provided in Fig. 11.

## B.3  PROMPT THEMES

To comprehensively assess the generative capabilities of T2I models across diverse scenarios, we design the benchmark prompts to achieve broad thematic coverage. As illustrated in Fig. 12, the prompts are organized into five major theme categories: *Art*, *Illustration*, *Creative Divergence*, *Design*, and *Film&Storytelling*, which are further divided into 20 subcategories. This hierarchical design ensures comprehensive coverage of practical application scenarios while enabling detailed evaluation across different creative domains. We provide several prompt cases of each prompt theme to facilitate understanding in Fig. 12.

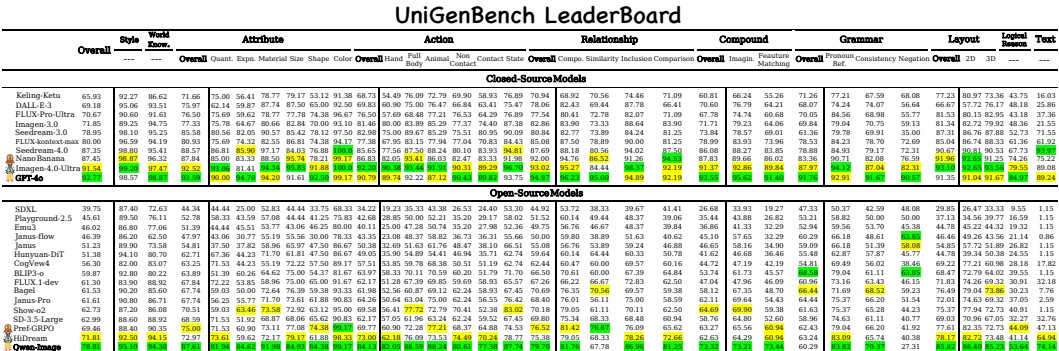

Figure 11: **Fine-grained Benchmarking Results of T2I models on UNIGENBENCH.** Best scores are in green, second-best in yellow.

## Prompt Themes of UniGenBench

Figure 12: **Prompt Themes of UNIGENBENCH.** We provide representative prompt examples for each theme to facilitate understanding.

## B.4 SUBJECT CATEGORIES

As shown in Fig. 3 (b), we further design the benchmark to cover a diverse range of subject categories, including *animals*, *objects*, *anthropomorphic characters*, and *scenes*. Moreover, an *Other* category is introduced to capture special entities that emerge in specific themes, such as robots in science-fiction themes or sculptures in artistic contexts, thereby ensuring that the benchmark reflects a broader spectrum of generation subjects.

## B.5 EVALUATION DIMENSIONS

With the rapid advancement of T2I models, their overall generative performance on mainstream evaluation dimensions, such as *object attributes* and *actions*, has already reached a relatively high level. We argue that future evaluations should move beyond these coarse dimensions and adopt a finer-grained decomposition, thereby more precisely identifying a model's strengths and weaknesses across specific sub-tasks and providing deeper insights into its true capabilities and limitations. To this end, our benchmark defines 10 primary evaluation dimensions, six of which are further decomposed into fine-grained sub-dimensions, as illustrated in Fig. 13. These include several critical aspects that are largely overlooked by existing benchmarks:

- **Logical Reasoning**: Evaluates a model's ability to handle prompts requiring causal, contrastive, or other simple logical inferences.

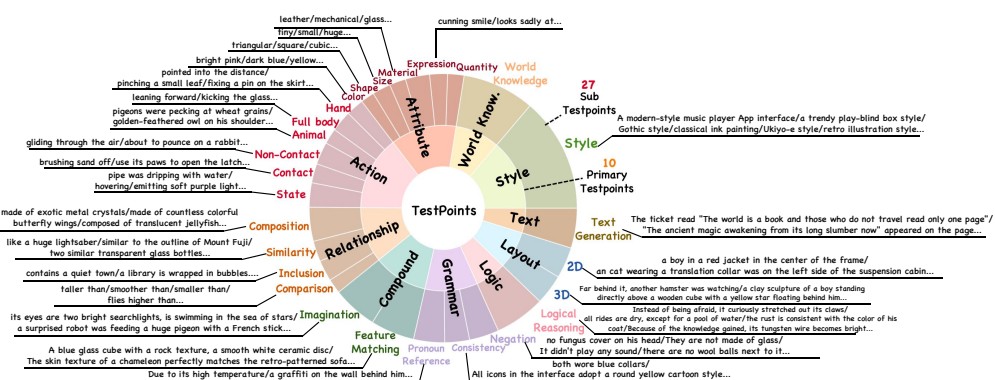

Figure 13: **Evaluation Dimensions of UNIGENBENCH**. We provide representative prompt examples for each evaluation dimension to facilitate understanding.

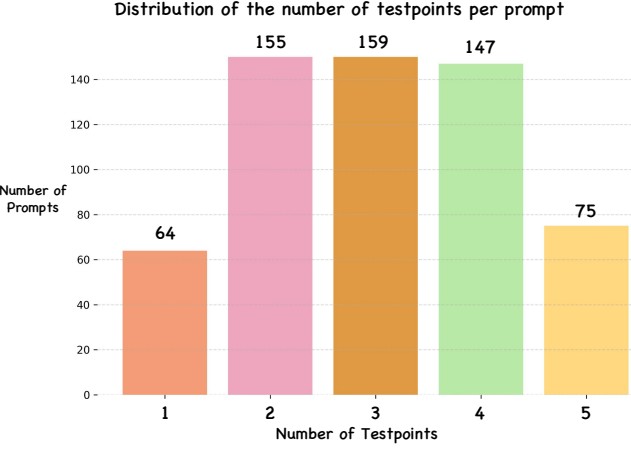

Figure 14: **Distribution of Testpoint Counts in Prompts**. This figure presents the distribution of the number of testpoints per prompt in UNIGENBENCH.

- **Facial Expressions**: Assesses whether generated characters exhibit correct and contextually appropriate emotions.

- **Pronoun Reference**: Tests the model's capability to resolve ambiguous pronouns (e.g., *its*, *him*) correctly.

- **Hand Actions**: Examines whether fine-grained hand movements and gestures are accurately rendered.

- **Composition Relations**: Measures understanding of "made of" or "composed of" relations among objects.

- **Similarity Relations**: Evaluates the ability to represent resemblance (*e.g.,* "two similar objects", "looks like...").

- **Inclusion Relations**: Tests comprehension of "contains" or "inside" relationships among entities.

- **Grammatical Consistency**: Assesses whether multiple objects correctly share attributes or features specified in the prompt (*e.g.,* "both red balloons...").

We believe that incorporating these fine-grained dimensions is essential for evaluating nuanced semantic comprehension and for ensuring closer alignment with human intent. We also provide several prompt cases of each evaluation dimension in Fig. 13.

### B.6 DISTRIBUTION OF TESTPOINT COUNTS IN PROMPTS

Unlike other benchmarks that contain thousands of prompts, UniGenBench only requires 600 prompts, each focusing on 1 to 5 specific testpoints, ensuring both breadth and efficiency in evaluation. We visualize the distribution of testpoint counts across prompts, as shown in Fig. 14.

### B.7 SUPERIORITY OF UNIGENBENCH

The superiority of UNIGENBENCH can be summarized as follows:

- **Comprehensive Dimension Evaluation**: It spans 10 primary dimensions and 27 sub-dimensions, offering a systematic and in-depth assessment of a model's capabilities across various aspects. To the best of our knowledge, this is the most comprehensive benchmark in terms of evaluation dimensions.

- **Rich Prompt Theme Coverage**: The benchmark includes 5 major prompt themes and 20 sub-themes, covering a wide array of generation scenarios, ranging from realistic to creative tasks. This ensures a comprehensive evaluation of the model's generative capabilities across various scenarios.

- **Efficient and Effective**: Unlike other benchmarks Wei et al. (2025); Huang et al. (2023) that require thousands of prompts, UniGenBench utilizes only 600 prompts, each focused on 1 to 5 specific testpoints, ensuring both breadth and efficiency in evaluation.

- **Reliable MLLM Evaluation**: Each prompt is paired with detailed testpoint descriptions that clarify how the testpoints are manifested in the prompt, enabling MLLM to perform precise assessments. Unlike other methods Wei et al. (2025); Huang et al. (2023), which often require multiple questions per sample for evaluation, our approach streamlines the process, improving efficiency without compromising accuracy.

## C ETHICAL STATEMENT

In this work, we affirm our commitment to ethical research practices and responsible innovation. To the best of our knowledge, this study does not involve any data, methodologies, or applications that raise ethical concerns. All experiments and analyses were conducted in compliance with established ethical guidelines, ensuring the integrity and transparency of our research process.

## D DECLARATION ON LLM USAGE

In this paper, we use LLMs only for minor language polishing.

