# OpenReview forum: "Pref-GRPO: Pairwise Preference Reward-based GRPO for Stable Text-to-Image Reinforcement Learning"
_ICLR.cc/2026/Conference — Submitted to ICLR 2026_

### Official Review · Reviewer_7okx · 2025-11-01

**Soundness:** 2
**Presentation:** 3
**Contribution:** 2
**Rating:** 4
**Confidence:** 4

**Summary:**

This paper introduces PREF-GRPO, a pairwise preference reward-based GRPO method for text-to-image generation, and UNIGENBENCH, a unified benchmark for fine-grained evaluation. The authors claim their method addresses reward hacking by shifting from reward score maximization to pairwise preference fitting. While the paper tackles an important problem in T2I reinforcement learning, there are significant concerns regarding the experimental setup and claims validation.

**Strengths:**

1. The paper identifies an important problem in T2I reinforcement learning - reward hacking caused by illusory advantage.
2. The introduction of UNIGENBENCH with fine-grained evaluation dimensions is valuable for the community.
3. The pairwise preference approach is conceptually interesting and aligns with human evaluation processes.

**Weaknesses:**

1. Unfair Baseline Comparisons: The experimental setup raises significant concerns:
- PREF-GRPO uses UnifiedReward-Think while baselines use UnifiedReward without the thinking mechanism, creating an unfair advantage
- HPS is outdated (the community now primarily uses HPSv2, ImageReward, and MPS)
2. Insufficient Evidence for Reward Hacking Mitigation: The paper fails to provide convincing evidence that PREF-GRPO actually alleviates reward hacking:
- No analysis of training dynamics beyond basic reward curves
- Missing monitoring metrics on established benchmarks like GenEval throughout training
- The comparison in Table 1 only shows final performance, not the stability of the training process
3. Limited Ablation Studies: The paper lacks sufficient ablation studies to understand the contribution of different components of PREF-GRPO.

**Questions:**

- Have you conducted any analysis to show that PREF-GRPO actually reduces reward hacking behaviors rather than just achieving better final scores?
- How does the computational cost of PREF-GRPO compare to baseline methods, especially as group size increases?

---

> ### Author Response · Authors · 2025-11-19
>
> ## Response to Weakness 1
>
> ### **(1) Regarding Comparison of UnifiedReward**
>
> ​	UnifiedReward has been recognized as the strongest open-source reward model for pointwise score-based image generation RL, as demonstrated in recent studies such as [1]. Therefore, we selected it as the representative point score–based baseline for comparison. Notably, UnifiedReward also supports pairwise preference ranking without thinking mechanism.
>
> ​	Therefore, to further address the concern of fairness, we have conducted an additional experiment using the UnifiedReward to perform pairwise ranking under the same Pref-GRPO training pipeline. The results show that even without the “thinking” mechanism, Pref-GRPO still achieves consistent and significant improvements in generative model performance.
>
> |                                                  | UniGenBench  |   T2I-Comp   |   GenEval    | UnifiedReward |  PickScore   |
> | :----------------------------------------------: | :----------: | :----------: | :----------: | :-----------: | :----------: |
> |                    FLUX.1.dev                    |    61.30     |    48.17     |    62.92     |     3.04      |    22.42     |
> |          w/ UnifiedReward (Point Score)          |    63.62     |    50.20     |    67.28     |     3.14      |    22.88     |
> |  w/ PreF-GRPO (**UnifiedReward for Pair Rank**)  | 67.28 | 50.88 | 69.35 |  3.22  | 22.95 |
> | w/ PreF-GRPO (UnifiedReward-Think for Pair Rank) |  **69.46**   |  **51.85**   |  **70.53**   |   **3.26**    |  **23.02**   |
>
> ### **(2) Regarding Comparison of HPSv2**
>
> ​	We appreciate your suggestion regarding the version of HPS used. In fact, our implementation already employs **HPSv2**, which aligns with current community practice. We will update the manuscript to explicitly indicate “HPSv2” in the revised version to avoid any ambiguity.
>
>
>
> [1] Delving into RL for Image Generation with CoT: A Study on DPO vs. GRPO. NeurIPS, 2025.

---

> ### Author Response · Authors · 2025-11-19
>
> ## Response to Weakness 2 & Question 1
>
> ​	Thank you for your valuable suggestion! The *reward hacking* phenomenon in image generation RL manifests as **reward scores increasing while image quality deteriorates**. As illustrated in Fig. 2 and Fig. 8, this work empirically observe this behavior in several point score–based GRPO methods:
>
> - **HPSv2-based GRPO** tends to produce **over-saturated colors** (Fig. 8);
> - **UnifiedReward-based GRPO** often leads to **unnaturally dark styles** (Fig. 2).
>
> ------
>
> ### Root Cause: *Illusory Advantage*
>
> ​	Our work identifies the *root cause* of reward hacking as the **illusory advantage**, which arises from **minimal reward score differences** between similar generated images. After group normalization in GRPO, these small differences are **amplified disproportionately**, causing the model to **over-optimize trivial reward fluctuations** and **destabilize** the training.
>
> ------
>
> ### Our Solution: Pref-GRPO
>
> ​	To mitigate this issue, we propose **Pref-GRPO**, which transforms the traditional **point score–maximization objective**
>  into a **pairwise preference fitting objective**.
>  	By converting absolute reward scores into **pairwise win-rates**, Pref-GRPO naturally **increases the reward variance** within a sample group, resulting in a **more discriminative and robust reward distribution** for advantage estimation.
>
> ------
>
> ### Experimental Evidence
>
> We provide multiple empirical results demonstrating that Pref-GRPO effectively mitigates reward hacking:
>
> 1. **Variance Amplification (Fig. 1 a):**
>
>    Pref-GRPO maintains a **larger reward variance** during training, thereby preventing the illusory advantage and stabilizing optimization.
>
> 2. **Stable Quality Optimization (Fig. 2):**
>
>    While point score–based GRPO quickly suffers from visual degradation, Pref-GRPO **continues improving image quality even with prolonged training**.
>
> 3. **Style Preservation (Figs. 6 & 10):**
>
>    Using point score–based rewards (HPSv2 or UnifiedReward) often collapses the generation style. In contrast, Pref-GRPO **preserves the model’s original visual style while improving semantic fidelity and realism**.
>
> 4. **Point score-based winrate using UnifiedReward (Sec. A.2 & Fig. 7 & Table 4):**
>
>    We also convert point score -based reward model’s absolute scores into pairwise win-rates to approximate pairwise preference fitting. **This simple transformation alleviates the “dark-style” reward hacking problem qualitatively and yields moderate quantitative improvement**. However, it remains significantly weaker than our Pref-GRPO，which using pairwise preference reward model, i.e., UnifiedReward-Think for  winrate reward, confirming that **explicit pairwise preference modeling** provides a more reliable and human-aligned learning signal than pointwise scoring.
>
> 5. **Monitoring metrics on UniGenBench, T2I-CompBench, and GenEval**
>
> ​	As suggested, we add **monitoring metrics on several benchmarks** to demonstrate our Pref-GRPO stable optimization throughout training.
>
> | Training Step           | UniGenBench | T2I-Comp  | GenEval   |
> | ----------------------- | ----------- | --------- | --------- |
> | **0 (FLUX.1.dev)**      | 61.30       | 48.17     | 62.92     |
> | **50**                  | 63.42       | 49.21     | 64.58     |
> | **100**                 | 65.77       | 50.02     | 66.11     |
> | **150**                 | 67.03       | 50.76     | 68.04     |
> | **300**                 | 68.52       | 51.42     | 69.37     |
> | **w/ Pref-GRPO (Ours)** | **69.46**   | **51.85** | **70.53** |
>
> ​	As shown in the table above, our Pref-GRPO **consistently improves performance across all benchmarks during training**. This stability indicates that our pairwise preference–based optimization effectively mitigates instability and promotes robust enhancement of T2I model.
>
> ---
>
> ​	Overall, we believe our experiments provide strong evidence that our method actually alleviates reward hacking, which is also recognized by Reviewer gvS1 and Reviewer mni3.

---

> ### Author Response · Authors · 2025-11-19
>
> ## Response to Weakness 3
>
> ​	Thank you for your valuable comment! Our main contribution lies in revealing that reward hacking fundamentally stems from the "illusory advantage" problem, and in transforming the RL optimization objective from pointwise reward-score maximization to pairwise preference fitting to effectively address it. **Since our approach focuses on an objective-level reformulation rather than introducing multiple additional components, the component-wise ablations is naturally limited.**
>
> ​	Nevertheless, we have conducted multiple targeted experiments to prove the effectiveness of our proposed pairwise preference fitting mechanism:
>
> ***
>
> 1. ### **Direct Comparison vs. Point Score Baselines (Fig. 1, Fig. 2, Fig. 6 and Table 2)**
>
> ​	We compare Pref-GRPO against multiple point score–based GRPO variants (e.g., HPSv2, UnifiedReward). The experiments  show consistent and significant improvements in semantic consistency, realism, and robustness.
>
> 2. ### **Explore Point Score-based Winrate (Sec. A.2)**
>
> ​	To demonstrate that shifting the training objective to our proposed pairwise preference fitting enhances stability and that pairwise comparisons are more reliable than pointwise scores, we conduct an experiment using point score-based win rates as rewards. As shown in Fig. 7 of our paper, when the training objective shifts to pairwise preference fitting, the previously dominant dark style in the images is notably alleviated, which validates that pairwise preference fitting stabilizes training.
>
> 3. ### **Joint Optimization (Pref-GRPO + Reward Score Maximization) (Table 4 & Fig. 9)**
>
> ​	We conduct joint optimization using our Pref-GRPO and CLIP. As shown in Tab. 4, the integration of CLIP notably improves semantic consistency, but this gain comes at the expense of slightly reduced image quality, highlighting a trade-off between semantic alignment and visual fidelity. We also provide qualitative comparison results in Fig. 9, where, despite the quality trade-off, no reward hacking phenomenon is observed. These results indicate that pairwise preference fitting acts as a regularizer when combined with reward score maximization, providing a principled way to balance semantic accuracy and visual quality while mitigating reward hacking.
>
>
>
> ## Response to Question 2
>
> ​	Thank you for raising this insightful concern! As suggested, we conduct a detailed analysis of  the actual **wall-clock training time**, and we report the empirical results below.
>
> ---
>
> ### **Experimental Details**
>
> ​	We train FLUX.1.dev using 4 × H800 GPUs and deploy the reward computing service using the vLLM [1] tool with 4 GPUs, which significantly improves reward inference throughput. We compare the reward computing speed of point score using UnifiedReward (the strongest baseline) and pairwise preference winrate (our method) under different group sizes.
>
> ---
>
> ### **Experimental Result Analysis**
>
> - As shown below, under the typical setting (group = 8), our pairwise winrate reward increases the per-step training time by only about **4 seconds** compared to the point score method.
> - When the group size increases to 16, although the number of reward computations becomes 7.5 times larger than that of the point score method, the training process is only **17 seconds** slower.
>
> ​	This efficiency largely benefits from the vLLM-based parallel inference architecture, which effectively mitigates the potential computational bottleneck.
>
> |                                    | Point Score | Point Score | Pairwise Winrate (Our Setting) | Pairwise Winrate | Pairwise Winrate |
> | :--------------------------------: | :---------: | :---------: | :----------------------------: | :--------------: | :--------------: |
> |     **Rollout number / group**     |      8      |     16      |               8                |        12        |        16        |
> | **Reward computing number / step** |     32      |     64      |              112               |       264        |       480        |
> |     **Computing speed / step**     |     3 s     |     5 s     |              7 s               |       14 s       |       22 s       |
>
> ​	Our work also demonstrates that **a group size of 8 is sufficient to achieve stable optimization**. Although this introduces a minor time overhead, our method achieves significant quantitative improvements and effectively alleviates reward hacking, as demonstrated in Table 2, Figure 2 and 6 of the paper.
>
> ​	We believe this trade-off between computation and performance to be reasonable and well-justified.
>
>
>
> [1] Efficient Memory Management for Large Language Model Serving with PagedAttention, Proceedings of the ACM SIGOPS 29th Symposium on Operating Systems Principles, 2023.

---

> ### Author Response · Authors · 2025-11-27
> **Kindly reminder**
>
> Dear Reviewer 7okx,
>
> We hope this message finds you well. We sincerely appreciate the time and effort you have invested in reviewing our paper.
>
> We have posted a detailed response addressing your comments and would like to kindly follow up to see if our clarifications have sufficiently resolved your concerns. We deeply value your opinion and would appreciate any feedback you might have before the discussion period ends.
>
> Please let us know if there are any remaining points you would like us to clarify.
>
> Best regards,
>
> Pref-GRPO Authors

---

### Official Review · Reviewer_mni3 · 2025-11-01

**Soundness:** 3
**Presentation:** 4
**Contribution:** 3
**Rating:** 6
**Confidence:** 3

**Summary:**

The paper presents a GRPO method for tuning diffusion models with rewards along with a comprehensive benchmark for evaluating text-to-image models. Compared to previous GRPO methods for diffusion/flow models (e.g. {Flow,Mix,Dance}-GRPO), the proposed approach (Pref-GRPO), does not directly compute a reward per sample, but rather computes the reward over the full batch. This appears to make the advantage computation more reliable, leading to better performance.
Additionally, the paper also introduces "UniGenBench", which has several categories for comprehensive evaluation is evaluated using the LLM-as-a-judge methodology with Gemini 2.5 Pro.

**Strengths:**

I think the paper proposes an interesting mechanism to deal with the issue of reward-hacking in the reward optimization of diffusion models, which is backed by fairly strong results on several benchmarks.

The proposed UniGenBench also appears to be a useful addition in terms of evaluating newer models. While I'm not entirely sure that this would become a widely used benchmark given that the field has several benchmarks already (GenEval, T2I-Compbench, DPG-Bench, GenAI-Bench, TIFA etc.), it does have its merits.

**Weaknesses:**

[Major]

Comparisons with Previous GRPO work: I think the most important question is regarding the performance of Pref-GRPO compared to other formulations (Flow-GRPO, DanceGRPO etc.). The key claim being made in the paper is that the pairwise reward formulation is better suited to compute advantages for optimizing the model compared to existing work. While there seem to be some results in Fig. 2, Tab. 4-6, I'm not exactly sure how these settings compare to the previous GRPO methods.


[Minor]

A slightly curious aspect of the paper is that the benchmark and mehtod are quite orthogonal; i.e the benchmark on its own can provide useful analysis, while the method could also be validated on existing benchmarks, and to some extent the paper feels like 2 disjoint works stitched together.

**Questions:**

The only question I'd really like to have full clarity is the differences and advantages over other GRPO frameworks for diffusion models. While I'm leaning to accept the paper, I think answering this comprehensively would be ideal.

---

> ### Author Response · Authors · 2025-11-19
>
> ## Response to Weakness 1 & Question 1
>
> ​	Thank you very much for this valuable and insightful comment.
>
> ### **(1) Comparisons to Previous GRPO Frameworks**
>
> ​	Both **DanceGRPO** and **Flow-GRPO** are concurrent works that adopt similar *point score–based reward optimization* for diffusion models. Their optimization objectives are designed to **maximize scalar reward scores** predicted by reward models (e.g., HPSv2, UnifiedReward). In our implementation, all **point score–based GRPO baselines** follow the training pipeline and hyperparameter settings of **DanceGRPO**, ensuring a fair and consistent comparison.
>
> ### **(2) Limitations of Traditional Point Score–based GRPO**
>
> ​	Through extensive experiments, we observe that point score–based GRPO methods can lead to **illustratry advantages**，induced by minimal reward score differences between generated images assigned by reward models, which results in reward hacking, where scores increase but image quality deteriorates. As illustrated in **Fig. 6** of our paper:
>
> - Using **HPSv2-based GRPO** tends to produce **over-saturated colors**;
> - Using **UnifiedReward-based GRPO** often results in **unnaturally dark styles**.
>
> ### **(3) Our solution: Pairwise preference Reward–based GRPO** (Pref-GRPO)
>
> ​	To address this, our proposed **Pref-GRPO** reformulates the objective from *reward score maximization* to *pairwise preference fitting*. Instead of optimizing absolute reward score, the model learns from pairwise preference-based winrate. This design provides three key advantages (as detailed in lines 283–294 of the paper):
>
> - **Amplified Reward Variance:**  By transforming absolute reward scores into pairwise win-rates, Pref-GRPO inherently increases reward variance across a group of generated images
> - **Robustness to Reward Noise:**  Because the optimization relies on relative rankings rather than raw scores, Pref-GRPO substantially mitigates the amplified impact of small reward score fluctuations or biases in the reward model.
> - **Better Alignment with Human Preference:**  The pairwise formulation mirrors human perceptual evaluation. When comparing two images of similar quality, human judgments are inherently relative rather than absolute. By emulating this process, Pref-GRPO captures fine-grained quality distinctions often missed by pointwise scoring, providing a more faithful and reliable signal for policy improvement.
>
> ### **(4) Experimental Results**
>
> ​	As shown in Fig. 2 and Fig. 6, Pref-GRPO produces **more semantically consistent and visually appearly images** compared to point score–based GRPO variants based on DanceGRPO. Furthermore, Table 1–3 demonstrate that our method achieves significant quantitative improvements across diverse human-aligned reward models and benchmarks.
>
> ### **(5) Additional Comparison Results with Baseline Trained on FlowGRPO**
>
> ​	To further highlight the effectiveness of our approach, we additionally trained the strongest baseline (**UnifiedReward**) under the **FlowGRPO** training framework. The results are summarized below:
>
> |                                  | UniGenBench | T2I-Comp  |  GenEval  | UnifiedReward | PickScore |
> | :------------------------------: | :---------: | :-------: | :-------: | :-----------: | :-------: |
> |            FLUX.1.dev            |    61.30    |   48.17   |   62.92   |     3.04      |   22.42   |
> | w/ UnifiedReward (**DanceGRPO**) |    63.62    |   50.20   |   67.28   |     3.14      |   22.88   |
> | w/ UnifiedReward (**FlowGRPO**)  |    64.16    |   49.89   |   67.44   |     3.17      |   22.91   |
> |     **w/ PreF-GRPO (Ours)**      |  **69.46**  | **51.85** | **70.53** |   **3.26**    | **23.02** |
>
> ​	Since DanceGRPO and FlowGRPO share a similar training pipeline and both adopt point score–based reward maximization, their performance is also quite close.
> ​	However, our PreF-GRPO achieves **consistent and substantial improvements** across all benchmarks including **UniGenBench**, **T2I-Comp**, and **GenEval** ,as well as under all human-aligned reward models (**UnifiedReward** and **PickScore**).
>
> ---
>
> ​	In summary, while Flow-GRPO and DanceGRPO maximize scalar rewards from specific reward models, **Pref-GRPO introduces a fundamentally different optimization objective based on pairwise human preference alignment**. This shift not only mitigates reward hacking but also ensures more stable, human-faithful optimization of T2I models.

---

> ### Author Response · Authors · 2025-11-19
>
> ## Response to Weakness 2
>
> ​	Thank you for this insightful comment! Actually, our UniGenBench and method (Pref-GRPO) are in fact closely connected and mutually reinforcing components of our work.
>
> ​	In developing Pref-GRPO, we identified a critical limitation in existing community benchmarks, they fail to provide fine-grained evaluation of current advanced text-to-image generation models. As shown in Table 5 of paper, for example, FLUX already achieves **97.81** in the *single-object* dimension of GenEval, indicating that existing benchmarks have become **saturated** and **lack the sensitivity** to capture nuanced improvements.
>
> ​	To address this gap, we constructed UniGenBench, a more comprehensive and human-aligned benchmark that enables **detailed and interpretable analysis** of generative model performance.
>
> |                 | Primary Dimension | Sub Dimension | Prompt Theme | Prompt Num. | Multi-Testpoint per Prompt |
> | :-------------: | :---------------: | :-----------: | :----------: | :---------: | :------------------------: |
> |     General     |         6         |       -       |      -       |     553     |             -              |
> | T2I-CompBench++ |         8         |       -       |      -       |    2400     |             -              |
> |    DPG-Bench    |         9         |       -       |      -       |    1065     |             -              |
> |      WISE       |         6         |       -       |      -       |    1000     |             -              |
> |   TIIF-Bench    |         9         |       -       |      -       |    5000     |            1~2             |
> | **UniGenBench** |        10         |      27       |      20      |     600     |            1~5             |
>
> ​	As shown in the table above, UniGenBench offers a more efficient yet comprehensive evaluation framework compared to existing benchmarks.  Through a multi-testpoint design per prompt, it requires only 600 prompts to achieve **broader primary-dimension coverage** and **fine-grained sub-dimension evaluation**.
>
> ​	Besides, UniGenBench is not only a standalone contribution but also **essential for accurately evaluating the improvements** brought by our **Pref-GRPO** method. In our experiments, we train models **on the training subset of UniGenBench** and evaluated them **in-domain** on UniGenBench itself. This setup allows us to more precisely measure how Pref-GRPO enhances semantic consistency and alignment with human preference.
>
> ---
>
> In summary, we believe the integration of UniGenBench and Pref-GRPO **is deliberate and necessary**:
>
> - UniGenBench provides a **more discriminative and interpretable evaluation environment**;
>
> - Pref-GRPO offers a **methodological advancement** that leverages such a benchmark to demonstrate concrete improvements.
>
> Together, they form a **coherent and complementary framework** that advances both **evaluation** and **optimization** for human-aligned T2I model optimization.

---

> ### Author Response · Authors · 2025-11-27
> **Kindly reminder**
>
> Dear Reviewer mni3,
>
> We hope this message finds you well. We sincerely appreciate the time and effort you have invested in reviewing our paper.
>
> We have posted a detailed response addressing your comments and would like to kindly follow up to see if our clarifications have sufficiently resolved your concerns. We deeply value your opinion and would appreciate any feedback you might have before the discussion period ends.
>
> Please let us know if there are any remaining points you would like us to clarify.
>
> Best regards,
>
> Pref-GRPO Authors

---

### Official Review · Reviewer_GeGc · 2025-11-01

**Soundness:** 3
**Presentation:** 3
**Contribution:** 2
**Rating:** 4
**Confidence:** 2

**Summary:**

This paper introduces PreF-GRPO, a pairwise preference reward-based Group Relative Policy Optimization (GRPO) method designed to address the reward hacking problem in text-to-image (T2I) reinforcement learning. By shifting from pointwise reward maximization to pairwise preference fitting, the method aims for more stable learning and better alignment with nuanced human preferences. Additionally, the authors present UniGenBench, a new benchmark for T2I generation that evaluates models across comprehensive primary. They also fine-grained sub-dimensions using an automated pipeline based on Multi-modal Large Language Models (MLLMs). Through extensive experiments, the authors demonstrate that PreF-GRPO achieves notable improvements in both image quality and semantic consistency compared to existing baselines.

**Strengths:**

[+] For the reward hacking problem in existing GRPO methods, this paper identifies “illusory advantage” resulting from minimal score differences and their amplification during normalization

[+] It is well motivated that replace pointwise rewards with pairwise preference-based win rates, which is illustrated in Figure 1.

[+] The description and motivation of UniGenBench as a benchmark is solid, with Figure 3 and Figure 4 substantiating its comprehensiveness.

**Weaknesses:**

[-] There is a lack of formal analysis on the illusory advantage phenomenon. It would be better to have a more rigorous analysis quantifying the expected change in variance between score-based and pairwise win-rate rewards.

[-] Most ablations focus on comparing PreF-GRPO to existing pointwise methods or naive adaptations, without the adversarial preference model errors analysis.

[-] The assertion that PreF-GRPO produces more “human-aligned” or “faithful” outputs is plausible but not decisively demonstrated without human assessment.

**Questions:**

1. Can the authors provide a more formal mathematical analysis of reward variance and its amplification?
1. How well does  MLLM-based automated evaluation (via Gemini2.5-pro) agree with human annotators?

---

> ### Author Response · Authors · 2025-11-19
>
> ## Response to Weakness 1 & Question 1
>
> ​	Thanks for your valuable suggestion! We provide **formal analysis of the Illusory Advantage Phenomenon** as follows.
>
> ---
>
> ### Preliminary
>
> ​	Group Relative Policy Optimization (GRPO) introduces a group-relative advantage to stabilize policy updates.
> For a group of $G$ generated images $\\{x _ 0^i\\}_{i=1}^G$, the advantage of the $i$-th image is defined as:
>
> $$
> \hat{A} _ t^i =
> \frac{R(x _ 0^i, c) - \mathrm{mean}(\{R(x _ 0^j, c)\} _ {j=1}^G)}{\mathrm{std}(\{R(x _ 0^j, c)\} _ {j=1}^G)}
> = \frac{\Delta r_i}{\sigma_r},
> $$
>
> where $\sigma_r$ is the standard deviation of the group reward scores.  This normalization ensures that the advantage reflects a sample’s relative performance within the group.
>
> The GRPO optimization objective for flow matching models can be expressed as:
>
> $$
> \mathcal{J} _ {\text{Flow-GRPO}}(\theta) =
> E _ {c, \\{x^i\\}} \Bigg[
> \frac{1}{G} \sum_{i=1}^G \frac{1}{T}\sum_{t=0}^{T-1}
> \min\big(r _ t^i(\theta) \hat{A} _ t^i, \text{clip}(r _ t^i(\theta), 1-\eta, 1+\eta)\hat{A} _ t^i \big) - \beta D_{\mathrm{KL}}(\pi_\theta \| \pi_{\text{ref}})
> \Bigg],
> $$
>
> where
>
> $$
> r_t^i(\theta) = \frac{p_\theta(x_{t-1}^i|x_t^i,c)}{p_{\theta_{\text{old}}}(x_{t-1}^i|x_t^i,c)}.
> $$
>
> ### Gradient Derivation
>
> ​	For clarity, we ignore the clip operation and the KL regularization in the following.  The gradient of the GRPO objective with respect to model parameters $\theta$ is:
>
> $$
> \nabla_\theta \mathcal{J} _ {\text{Flow-GRPO}}(\theta)
> = E _ {c, \\{x^i\\}}
> \Bigg[
> \frac{1}{G} \sum_{i=1}^G \frac{1}{T}\sum_{t=0}^{T-1}
> \nabla_\theta \big( r _ t^i(\theta) \hat{A} _ t^i \big)
> \Bigg].
> $$
>
> Substituting $\hat{A}_t^i = \Delta r_i / \sigma_r$, we obtain:
>
> $$
> \nabla_\theta \mathcal{J} _ {\text{Flow-GRPO}}(\theta)
> = \frac{1}{\sigma_r} \,
> E _ {c, \\{x^i\\}}
> \Bigg[
> \frac{1}{G} \sum_{i=1}^G \frac{1}{T}\sum_{t=0}^{T-1}
> \nabla_\theta \big( r _ t^i(\theta) \Delta r _ i \big)
> \Bigg].
> $$
>
> Hence, the gradient magnitude is **inversely proportional to the reward variance** $\sigma_r$:
>
> $$
> \|\nabla_\theta \mathcal{J}_{\text{Flow-GRPO}}\| \propto \frac{1}{\sigma_r}.
> $$
>
> ### Interpretation: *Illusory Advantage*
>
> ​	When current point score-based reward models assign overly similar scores to images within the same group (i.e.,
> $R(x _ 0^i, c) \approx R(x _ 0^j, c)$), the standard deviation $\sigma_r \to 0$.  In this case, even small differences $\Delta r _ i$ are **amplified** by $1/\sigma_r$, producing excessively large normalized advantages $\hat{A} _ t^i$ and consequently **unstable gradient updates**.
>
> ​	This over-amplification causes the model to **over-optimize** for minor or spurious reward differences, which can manifest as *reward hacking*.  We term this instability the ***illusory advantage*** phenomenon. Formally, as $\sigma_r \to 0$:
>
> $$
> \lim_{\sigma_r \to 0} \nabla_\theta \mathcal{J}_{\text{Flow-GRPO}}(\theta) \to \infty,
> $$
>
> indicating overly aggressive policy updates and loss of optimization stability.
>
> ​	As shown in Fig. 1 (a) of our paper, **Pref-GRPO** provides amplified reward variance, since high-quality samples are pushed toward win-rates near 1, while lower-quality samples approach 0.  This produces a reward distribution that is both more discriminative and more robust for advantage estimation, effectively reducing reward hacking and stabilizing policy optimization.

---

> ### Author Response · Authors · 2025-11-19
>
> ## Response to Weakness 2
>
> ​	Thank you for this valuable suggestion! As suggested, we conduct additional experiments to examine the robustness of Pref-GRPO against potential **adversarial preference noise**.
>
> ---
>
> ### **Experimental Details**
>
> ​	Specifically, during training, we introduce **10% random noise** into the pairwise preference reward：that is, for each image pair, **there was a** **0.1 probability that the preference result was randomly flipped**.
>
> ### **Experimental Result Analysis**
>
> ​	Experimental results are summarized below. Despite the deliberate injection of noise, **Pref-GRPO still maintains strong performance** across all benchmarks compared with traditional point score-based method, with only marginal degradation compared to the clean setting. This demonstrates that our proposed pairwise preference optimization is **robust to imperfect or noisy preference signals**, which often occur in real-world feedback scenarios.
>
> |                                     | UniGenBench  | T2I-Comp     | GenEval      | UnifiedReward | PickScore    |
> | ----------------------------------- | ------------ | ------------ | ------------ | ------------- | ------------ |
> | FLUX.1.dev                          | 61.30        | 48.17        | 62.92        | 3.04          | 22.42        |
> | w/ HPS+CLIP                         | 58.77        | 46.77        | 59.31        | 3.09          | 22.62        |
> | w/ UnifiedReward                    | 63.62        | 50.20        | 67.28        | 3.14          | 22.88        |
> | w/ Pref-GRPO (10% **reward noise**) | 67.92 | 51.03 | 70.12 | 3.22   | 22.90 |
> | w/ Pref-GRPO (**Ours**)             | **69.46**    | **51.85**    | **70.53**    | **3.26**      | **23.02**    |
>
>
>
> ## Response to Weakness 3
>
> We appreciate the reviewer’s concern about the evidence for human alignment.
>
> ### **Human-Preference-Aligned Reward Model Evaluation.**
>
> ​	In this work, we employ widely adopted human preference reward models from the community, which **have been explicitly aligned with large-scale human preference datasets** to evaluate image quality, such as UnifiedReward, PickScore and ImageReward. As shown in Table 2 of paper, our PreF-GRPO consistently outperforms the traditional point score–based methods across all human-aligned reward models, indicating that the improvements are robust under multiple human-preference-oriented evaluation metrics.
>
> ### **Human Study**
>
> ​	As suggested, we conduct an additional human study to further validate the alignment of our method with human preferences. Specifically, we invite five trained annotators to perform **pairwise ranking comparisons** between images generated by the strongest baseline (FLUX w/ UnifiedReward) and our Pref-GRPO on 150 prompts sampled from UniGenBench. The annotators evaluate the results across two dimensions: semantic consistency, and image quality.
>
> The win rates are summarized below, where ***Tie*** indicates that the two images have comparable quality.
>
> |                      | FLUX w/ UnifiedReward | FLUX w/ PreF-GRPO |  Tie  |
> | :------------------: | :-------------------: | :---------------: | :---: |
> | Semantic Consistency |         24.4%         |     **62.7%**     | 12.9% |
> |    Image Quality     |         11.4%         |     **87.4%**     | 1.2%  |
>
> ​	As shown in the table, our Pref-GRPO demonstrates clear superiority across all evaluation dimensions, achieving notably higher win rates in terms of semantic consistency, and image quality.
>
> ​	These results provide direct human evidence that Pref-GRPO generates outputs that are more human-aligned, faithful, and visually appealing.
>
>
>
> ## Response to Question 2
>
> ​	Thank you for your concern about the reliablity of our MLLM-based evaluation. As shown in Fig. 5(b), in our benchmark evaluation, each prompt covers multiple test points, and each test point is explicitly annotated to indicate how it is reflected in the prompt. This design helps the MLLM (Gemini-2.5-Pro) perform reliable and interpretable evaluations rather than relying on holistic subjective judgment.
>
> **Human review**
>
> ​	To verify its accuracy, we conduct a **human review study**. Specifically, three trained annotators examined both the scores and rationale explanations produced by Gemini-2.5-Pro for 600 images generated by GPT-4o, Qwen-Image, and FLUX.1-dev. For each test sample, if the MLLM’s **evaluation across all assessment dimensions was correct, it was counted as accurate; otherwise, it was marked as incorrect.** The results are summarized below:
>
> |                | FLUX.1.dev | Qwen-Image | GPT-4o |
> | -------------- | ---------- | ---------- | ------ |
> | Gemini-2.5-Pro | 96.9%      | 96.3%      | 97.6%  |
>
> ​	These results demonstrate that Gemini-2.5-Pro achieves **high agreement (≈97%) with human annotators**, confirming that the MLLM-based evaluation is highly reliable and consistent with expert human judgment.

---

> ### Author Response · Authors · 2025-11-27
> **Kindly reminder**
>
> Dear Reviewer GeGc,
>
> We hope this message finds you well. We sincerely appreciate the time and effort you have invested in reviewing our paper.
>
> We have posted a detailed response addressing your comments and would like to kindly follow up to see if our clarifications have sufficiently resolved your concerns. We deeply value your opinion and would appreciate any feedback you might have before the discussion period ends.
>
> Please let us know if there are any remaining points you would like us to clarify.
>
> Best regards,
>
> Pref-GRPO Authors

---

### Official Review · Reviewer_gvS1 · 2025-11-11

**Soundness:** 3
**Presentation:** 3
**Contribution:** 3
**Rating:** 4
**Confidence:** 4

**Summary:**

This paper brilliantly identifies and solves "reward hacking" in T2I models. It argues the problem is "illusory advantage": standard reward models give similar images very similar scores, which, when normalized, create huge, noisy, and fake "advantages" that make training unstable.

The fix, PREF-GRPO, is elegant. Instead of using flawed absolute scores, it makes images in a group compete. It uses a pairwise model ("A is better than B") and gives each image a "win rate." This signal is far more stable and stops the model from hacking. As a huge bonus, the paper also introduces UNIGENBENCH, a new, super-detailed benchmark for T2I evaluation.

**Strengths:**

- The "illusory advantage" diagnosis is brilliant.

- The "win rate" solution is an elegant and effective fix.

- The visual proof is undeniable; the qualitative images show this method works and the others don't.

- It also contributes a fantastic new benchmark.

**Weaknesses:**

The main potential issue is computational cost. A "win rate" for a group of 8 images requires 28 pairwise comparisons ($O(G^2)$), versus just 8 for the baseline ($O(G)$). This seems significantly slower per training step, which the paper doesn't heavily focus on.

**Questions:**

How much does the $O(G^2)$ complexity of pairwise comparisons slow down the actual wall-clock training time compared to the $O(G)$ baseline?

---

> ### Author Response · Authors · 2025-11-19
>
> ## Response to Weakness 1 & Question 1
>
> ​	Thank you for raising this insightful concern! As suggested, we conduct a detailed analysis of  the actual **wall-clock training time**, and we report the empirical results below.
>
> ---
>
> ### **Experimental details**
>
> ​	We train FLUX.1.dev using 4 × H800 GPUs and deploy the reward computing service using the vLLM [1] tool with 4 GPUs, which significantly improves reward inference throughput. We compare the reward computing speed of point score using UnifiedReward (the strongest baseline) and pairwise preference winrate (our method) under different group sizes.
>
> ---
>
> ### **Experimental result analysis**
>
> - As shown below, under the typical setting (group = 8), our pairwise winrate reward increases the per-step training time by only about **4 seconds** compared to the point score method.
> - When the group size increases to 16, although the number of reward computations becomes 7.5 times larger than that of the point score method, the training process is only **17 seconds** slower per-step.
>
> ​	This efficiency largely benefits from the vLLM-based parallel inference architecture, which effectively mitigates the potential computational bottleneck brought by *O(G²)* comparisons.
>
> |                                    | Point Score | Point Score | Pairwise Winrate (Our Setting) | Pairwise Winrate | Pairwise Winrate |
> | :--------------------------------: | :---------: | :---------: | :----------------------------: | :--------------: | :--------------: |
> |     **Rollout number / group**     |      8      |     16      |               8                |        12        |        16        |
> | **Reward computing number / step** |     32      |     64      |              112               |       264        |       480        |
> |     **Computing speed / step**     |     3 s     |     5 s     |              7 s               |       14 s       |       22 s       |
>
> ​	Our work also demonstrates that **a group size of 8 is sufficient to achieve stable optimization**. Although this introduces a minor time overhead, our method achieves significant quantitative improvements and effectively alleviates reward hacking, as demonstrated in Table 2, Figure 2 and 6 of the paper.
>
> ​	We believe this trade-off between computation and performance to be reasonable and well-justified.
>
>
>
> [1] Efficient Memory Management for Large Language Model Serving with PagedAttention, Proceedings of the ACM SIGOPS 29th Symposium on Operating Systems Principles, 2023.

---

> ### Author Response · Authors · 2025-11-27
> **Kindly reminder**
>
> Dear Reviewer gvS1,
>
> We hope this message finds you well. We sincerely appreciate the time and effort you have invested in reviewing our paper.
>
> We have posted a detailed response addressing your comments and would like to kindly follow up to see if our clarifications have sufficiently resolved your concerns. We deeply value your opinion and would appreciate any feedback you might have before the discussion period ends.
>
> Please let us know if there are any remaining points you would like us to clarify.
>
> Best regards,
>
> Pref-GRPO Authors

---

### Meta-Review · Area_Chair_X9BV · 2026-01-02

**Summary:**

This paper works on preference learning for text-to-image generation. Authors proposed PREF-GRPO, pairwise preference reward-based GRPO method for T2I generation, establishing a more stable training. In each step, images within a generated group are pairwise compared using preference RM, and their win rate is calculated as the reward signal for policy optimization. Authors also proposed UNIGENBENCH, a unified T2I generation benchmark. Extensive experiments demonstrated the effectiveness of the proposed methods.

Before rebuttal, this paper got three 4 ratings and one 6 rating.

The strength of this paper given by reviewers are:
1. The "illusory advantage" diagnosis is great. (Reviewer gvS1, GeGc, 7okx)
2. The "win rate" solution is well motivated elegant, effective fix. (Reviewer gvS1, GeGc, 7okx)
3. The visual proof is undeniable; the qualitative images show this method works and the others don't. (Reviewer gvS1)
4. new benchmark is solid. (Reviewer gvS1, GeGc, mni3, 7okx)
5. proposes an interesting mechanism (Reviewer mni3)

The weakness & questions of this paper given by reviewers are:
1. The main potential issue is computational cost. (Reviewer gvS1)
2. a lack of formal analysis on the illusory advantage phenomenon. (Reviewer GeGc)
3. no adversarial preference model errors analysis. (Reviewer GeGc)
4. not decisively demonstrated without human assessment. (Reviewer GeGc)
5. Comparisons with Previous GRPO work. (Reviewer mni3)
6. the benchmark and method are quite orthogonal. (Reviewer mni3)
7. Unfair Baseline Comparisons. (Reviewer 7okx)
8. Insufficient Evidence for Reward Hacking Mitigation. (Reviewer 7okx)
9. Limited Ablation Studies. (Reviewer 7okx)

AC checked authors' paper, reviewers' comments and authors' rebuttal carefully and found authors addressed many concerns from reviewers. But still some concerns are not well addressed as detailed in the below session. Given these AC decided to reject this paper. But hope authors will find reviewers' comments are useful for their future research.

**Reviewer Concerns:**

weakness 1. authors mentioned under the typical setting (group = 8), increases the per-step training time by only about 4 seconds compared to the point score method, When the group size increases to 16, the training process is only 17 seconds slower per-step. but it is not clear how long each step take. so not sure the percentage added.

weakness 2. authors provided formal analysis on the illusory advantage phenomenon.

weakness 3. authors conducted additional experiments show the robustness of Pref-GRPO against potential adversarial preference noise.

weakness 4. authors provided user studies show their results is better.

weakness 5. authors clarified the different between this work and previous GRPO work. AC found that the win rate idea is quite similar to CaPO [1] paper. But that paper is not cited or compared. Though authors' work is an improvement over GRPO, discussion or compare DPO based method like CaPO is still important given the shared win rate idea.

[1] Lee, Kyungmin, et al. "Calibrated multi-preference optimization for aligning diffusion models." Proceedings of the Computer Vision and Pattern Recognition Conference. 2025.

weakness 6. authors mentioned that two parts are closely connected.

weakness 7. authors conducted more experiments to verify their algorithms still effective.

weakness 8. authors still didn't provide convincing evidence that PREF-GRPO actually alleviates reward hacking in the rebuttal. the table shows training step to 300. but what happened if training longer.

weakness 9. authors restate all the ablation studies done in the paper.

**Reviewer Scores:**

Reviewer gvS1 will keep their score 4.

Reviewer GeGc's comments are addressed. so may increase their score from 4 to something.

Reviewer mni3 might keep their score 6.

Reviewer 7okx might keep their score 4.

---

### Decision · Program_Chairs · 2026-01-26

Reject